# Laminin-defined mechanical status modulates retinal pigment epithelium phagocytosis

Aleksandra N Kozyrina [1,2,7], Teodora Piskova [1,2,7], Francesca Semeraro[1,2], Iris C Doolaar [3,4], Taspia Prapty[1,2], Tamás Haraszti [3,4], Maxime Hubert[5,6], Reinhard Windoffer[2], Rudolf E Leube [2], Ana-Sunčana Smith[5,6] & Jacopo Di Russo [1,2,3 ✉]

## Abstract

**Epithelial cells exhibit strong interconnections that are crucial for tissue mechanical properties. In homeostasis, these properties, termed mechanical homeostasis, depend on the balance between intercellular tension and extracellular matrix (ECM) adhesion forces. While age-related ECM remodeling is linked to outer retinal disease, its fundamental role in mechanical homeostasis remains unclear. In our study, we quantified changes in the mechanical state of retinal pigment epithelium (RPE), revealing a correlation with gradients of basement membrane laminins and their integrin receptors, β1 and β4. This relationship is related to regional phagocytic demand for recycling photoreceptor outer segments. Using a reductionist approach, we found that laminin 332 and laminin 511 isoforms differentially influence engagement with β1 and β4 integrins at low densities. Notably, laminin 511 enhances RPE contractility by reducing the β4 to β1 integrin engagement ratio, which subsequently diminishes phagocytic efficiency. Our findings suggest that the ECM-defined mechanical status of RPE serves as a novel parameter for visual function.**

**Keywords** Laminin; ECM; Mechanical Homeostasis; RPE
**Subject Categories** Cell Adhesion, Polarity & Cytoskeleton; Membranes & Trafficking

## Introduction

The mechanical properties of any tissue define its function and state (Anlaş and Nelson, 2018; Ayad et al, 2019). In a homeostatic state, these mechanical features are maintained in a preferred range for optimal activity, also known as tissue mechanical homeostasis (Eichinger et al, 2021). One of the main players in defining the mechanical homeostasis of tissues is its extracellular matrix (ECM) composition (Humphrey et al, 2014; Kozyrina et al, 2020).

In the retina, the most prominent ECM structure is the Bruch's membrane, situated at its outermost part, also known as outer retina (Booij et al, 2010). The Bruch's membrane enables nutrient exchange between the choroid vessels and the retina, and distally, the basement membrane layer provides structural support and adhesion to the retinal pigment epithelium (RPE). The RPE is a postmitotic cell monolayer which directly interfaces with the light-detecting neural tissue. In the developing eye, the RPE organizes the retina (Raymond and Jackson, 1995; German et al, 2008), while in the adult, it plays a key role in the visual cycle (Bok, 1993; Strauss, 2005). One of the main activities of RPE cells is the daily phagocytosis of photoreceptor cell fragments, therefore mediating the constant renewal of their light-sensitive outer segments and ensuring photoreceptor lifespan and light sensitivity (Lakkaraju et al, 2020; Young and Bok, 1969; Wald and Brown, 1956). The number of photoreceptor outer segments (POS) per RPE cell is not constant within the eye, with an average of 25–30 POS per RPE cell in the central and macular area and decreasing towards the periphery (~15 POS) (Volland et al, 2015), indicating a gradient of functional demand.

RPE cells are polygonal in shape and tightly adhere to each other: they arrange as a honeycomb-like structure in the centre, which gradually shifts to a more elongated and less organized structure towards the retinal periphery, which suggests a change in its mechanical status (Bhatia et al, 2016; Kim et al, 2021; Ortolan et al, 2022).

Although the Bruch's membrane is known to spatially vary along the visual axis (Booij et al, 2010) and undergo significant changes in aging and pathology (Piskova et al, 2023; Booij et al, 2010), it remains an open question how ECM cues fundamentally influence RPE mechanical homeostasis. Here, we hypothesize that the spatial variation of Bruch's membrane cues controls RPE mechanical status and region-dependent functional capacity.

In this work, we find that the relative density of the basement membrane laminin 511 and 332 defines the level of RPE contractility, which directly controls its functional efficiency. At low density, laminin 511 promotes high contractility in RPE, which differs from laminin 332 due to a lower β4-to-β1 integrin engagement. In vivo, we find that the density gradient of these

[1]Interdisciplinary Centre for Clinical Research, RWTH Aachen University, Pauwelstrasse 30, 52074 Aachen, Germany. [2]Institute of Molecular and Cellular Anatomy, RWTH Aachen University, Wendlingweg 2, 52074 Aachen, Germany. [3]DWI – Leibniz-Institute for Interactive Materials, Forckenbeckstrasse 50, 52074 Aachen, Germany. [4]Institute for Technical and Macromolecular Chemistry, RWTH Aachen University, Worringerweg 1-2, Aachen D-52074, Germany. [5]PULS Group, Department of Physics and Interdisciplinary Center for Nanostructured Films, Friedrich-Alexander University of Erlangen-Nürnberg, 91058 Erlangen, Germany. [6]Group for Computational Life Sciences, Division of Physical Chemistry, Rudjer Bošković Institute, 10000 Zagreb, Croatia. [7]These authors contributed equally: Aleksandra N Kozyrina, Teodora Piskova. ✉E-mail: jdirusso@ukaachen.de

laminin and integrin isoforms corresponds to changes in RPE mechanical status and functional demand.

These findings, for the first time, underline the importance of laminin-defined mechanical homeostasis in RPE and, in general, in the outer retina. The significance of RPE mechanical status for visual function in adults opens the question of its role in aging and sight-threatening diseases such as (high) myopia and age-related macular degeneration. Finally, this study could incentivize a larger use of Rho kinase inhibitors, currently used as anti-glaucoma medications, for other retinal diseases.

# Results

## Mechanical status of the RPE correlates with biochemical heterogeneity of Bruch's membrane

To understand the possible relation between RPE mechanical status and Bruch's membrane biochemical and mechanical cues, we characterized the outer retina of adult mice (25–30 weeks) (Figs. 1 and EV1). Bruch's membrane, retinal complexity and function have been described in relation to the visual axis (Volland et al, 2015; Ortolan et al, 2022). In both mice and humans, the central region (macula in humans) exhibits the highest density of photoreceptor cells and the highest photoreceptor to RPE ratio. This ratio gradually decreases along the visual axis, signifying a reduction in RPE functional demand (Volland et al, 2015). Hence, as previously shown (Bhatia et al, 2016; Kim et al, 2021; Ortolan et al, 2022), we identified three regions with respect to the radial distance from the optic nerve: centre (300–1200 µm), mid periphery (1200–2000 µm) and far periphery (2000–3000 µm) (Fig. 1A). We observed a substantial difference in the organization of RPE cells along the radial distance (Fig. 1A), consistent with previous reports (Bhatia et al, 2016; Kim et al, 2021; Ortolan et al, 2022). From a classic honeycomb arrangement in the centre, the RPE gradually changes to a monolayer consisting of more elongated cells at the far periphery (Figs. 1A and EV1A,D) (Bhatia et al, 2016; Kim et al, 2021; Ortolan et al, 2022).

The topology and morphology of an epithelial monolayer define its mechanical properties (Hannezo et al, 2014; Alt et al, 2017; Kaliman et al, 2021). Hence, we performed morphometric characterization of RPE cells in the different regions of the retina (Figs. 1B–D and EV1A–H). RPE cells have the largest area and perimeter in the central region ($\langle A \rangle = 469\ \mu m^2$; $\langle P \rangle = 84\ \mu m$), whereas they become smaller towards the mid periphery ($\langle A \rangle = 349\ \mu m^2$; $\langle P \rangle = 74\ \mu m$) and far periphery ($\langle A \rangle = 343\ \mu m^2$; $\langle P \rangle = 75\ \mu m$) (Fig. EV1D,E), aligning with previously published data (Bhatia et al, 2016; Kim et al, 2021; Ortolan et al, 2022). Next, to better quantify the mechanics of this differential cell arrangement, we calculated the shape factor for each cell, defined as the cell perimeter P divided by the square root of the corresponding cell area A. This density-independent shape factor has been theorized to describe epithelial rigidity based on the balance between cortical tension and cell-cell adhesion energy, with lower values where the first element is more prominent (Bi et al, 2015). Our data show an average value in the retinal central region of 3.8, which gradually increases until reaching values over 4.0 at the far periphery. This implies a gradual increase of intercellular tension, coinciding with the decreasing RPE functional demand (Figs. 1B,C and EV1F,G) (Volland et al, 2015). In addition, the comparison of the cellular

nuclear geometry shows a reduced roundness and circularity corresponding to a higher aspect ratio towards the far periphery, supporting the existence of different levels of intracellular strain (Fig. EV1H–J). Finally, we noticed the presence of two distinct cell populations in the centre and mid periphery, visible when cell area and cell perimeter distributions are normalized and compared (Figs. 1D and EV1K). Hence, we compared the topological relations of RPE cells across retinal regions by plotting the rescaled cell areas and perimeters as a function of the number of neighbors (respectively, Lewis' law (Lewis, 1931, 1928; Chiu, 1995; Kim et al, 2014) and Desh's law (Rivier, 2006)) (Figs. 1E and EV1L). Both comparisons show different curve behavior in the retinal regions, especially evident between the far periphery and other regions.

To gain further insight into monolayer topology, we then correlated the number of neighbors $n$ of each cell and the average number of neighbors $m(n)$ of any adjacent cells to the first with $n$ sides (Aboav-Weaire's laws) (Fig. 1F) (Aboav, 1980, 1970; Weaire, 1974). This relation describes the mean number of neighbors that neighboring cells have via the topological parameter $b(n)$ (see Methods section). The different regions show a linear relationship, as in any functional epithelium (Aboav, 1980, 1970; Weaire, 1974; Kaliman et al, 2021). Nevertheless, a shift is visible between values corresponding to retina regions, suggesting a topological variation of RPE from the centre to the far periphery (Figs. 1F and EV1M). Altogether, the extensive image-based morphometric and topological analyses (of more than 5500 cells from three mice—see Methods section) strongly support the existence of different mechanical status in the monolayer from the retinal central region to the far periphery.

To determine whether this mechanical gradient is influenced by environmental cues, we next quantified the distribution of mechanical and biochemical cues stemming from the Bruch's membrane. In the epithelial ECM, mechanical cues are mostly determined by the composition and crosslinking density of the interstitial matrix (Kozyrina et al, 2020). Therefore, we used confocal microscopy and immunofluorescence staining to quantify the relative amount of collagen type I and elastin present in the Bruch's membrane along the different regions (Fig. EV1N). Preparation artefacts like tissue tears and cavities were excluded to ensure data quality, using basal F-actin as a reference for the basal Z-position and analyzing signals only within manually defined ROIs for accuracy (Fig. EV1N, "stars"). The average fluorescence intensity quantification of the retina's central, mid peripheral, and far peripheral regions were comparable with the data reported in the literature (Chong et al, 2005; Newsome et al, 1987) (Fig. 1G). Furthermore, we measured the Young's modulus of the Bruch's membrane in different regions using atomic force microscopy (AFM) and indenting into the decellularised ECM by ~200 nm. The quantification resulted in a highly variable distribution of the data, with the centre having the most frequent (mode) stiffness of 5.5 kPa, the mid periphery 1.5 kPa and the far periphery 4 kPa (Fig. 1H). This biphasic stiffness trend suggests that ECM mechanical cues may not be the primary factor governing RPE mechanical homeostasis concerning functional demand.

From the biochemical point of view, the basement membrane regulates epithelial homeostasis through laminins—heterotrimeric proteins composed of α, β, and γ chains, that form distinct isoforms and play a critical role in cellular function (Kozyrina et al, 2020). Thus, we characterized the composition and relative amount of the

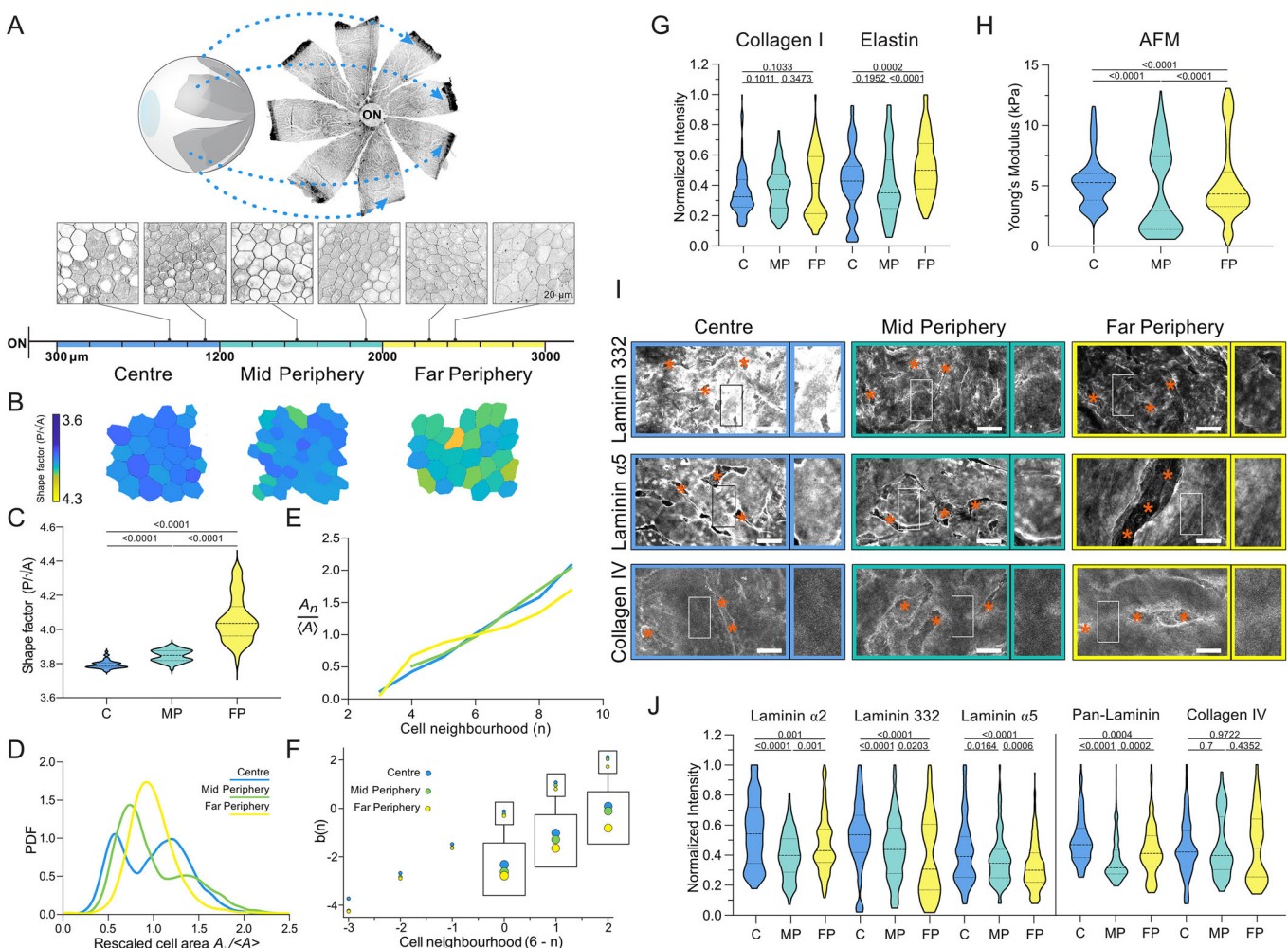

**Figure 1. Spatial heterogeneity of murine retinal pigment epithelium topology and Bruch's membrane composition.**

(A) Murine retina flat mount indicating the regions used for analyses and representative F-actin staining of RPE. The regions were defined by the distance from the optic nerve head: centre (300–1200 μm), mid periphery (1200–2000 μm) and far periphery (2000–3000 μm). (B) Representative cell shape index maps for the three regions. The shape index is identified as the perimeter divided by the square root of the area for each cell. (C) Violin plot of mean RPE shape factors from the different regions of the eye analyzed from 7 mice (biological replicates). (D) The probability density function (PDF) of the rescaled cell areas defined by $A_i/\langle A \rangle$, $A_i$ being the individual cell area and $\langle A \rangle$ being the average cell area within a given image. More than 5500 cells (technical replicates) were analyzed from 3 mice (biological replicates). (E) Lewis' law representing the relationship between rescaled cell area and neighborhood varying for three different regions from 3 mice (biological replicates). (F) Aboav-Weaire's laws showing the variation of the topological parameter b(n) depending on the number of neighbors for each neighboring cell between different retinal regions from 3 mice (biological replicates). (G) Violin plots of mean fluorescence intensity of interstitial matrix components of the Bruch's membrane. Data were collected from 3 mice (biological replicates), each with at least 10 technical replicates, and normalized to the highest intensity per mouse. (H) Violin plots of the regional Young's modulus of the Bruch's membrane from 5 mice (biological replicates). (I) Representative immunofluorescent images of the Bruch's membrane stained for laminin 332, laminin α5 and collagen type IV coupled with magnified view of quantified regions. Stars indicate preparation artefacts. Scale bar 20 μm and 10 μm for main images and magnifications, respectively. (J) Violin plots of mean fluorescence intensity of basement membrane components: laminin α2, laminin 332, laminin α5, pan-laminin, and collagen type IV. Each region contains the quantification of at least 7 sectors (technical replicates) from a minimum of 3 to 6 mice (biological replicates). The data (C, G, H, J) were statistically tested with Mann-Whitney test; exact p-values are indicated on the respective comparisons. Source data are available online for this figure.

main basement membrane components in the Bruch's membrane, namely laminins and collagen type IV (Figs. 1I,J and EV1N). Notably, we detected a significant difference in the relative amount of laminin isoforms along the visual axis, whereas collagen type IV remained unchanged. This supports the idea of the importance of laminins in the differential biochemical signaling to the RPE. Among the laminin isoforms detected in Bruch's membrane, we noticed a biphasic trend for laminin α2-containing isoforms, with higher (fluorescence) intensity in the center and far periphery and

lower in the mid periphery. Laminin 332 and isoforms containing laminin α5 had the highest (fluorescence) intensity in the centre that gradually diminished towards the periphery (Fig. 1I,J), following photoreceptor density and functional demand. No laminin α1- or α4-containing isoforms were detected.

Altogether, the characterization of the murine outer retina suggests a possible relationship between the relative density of Bruch's membrane's laminins and region-dependent RPE mechanical status.

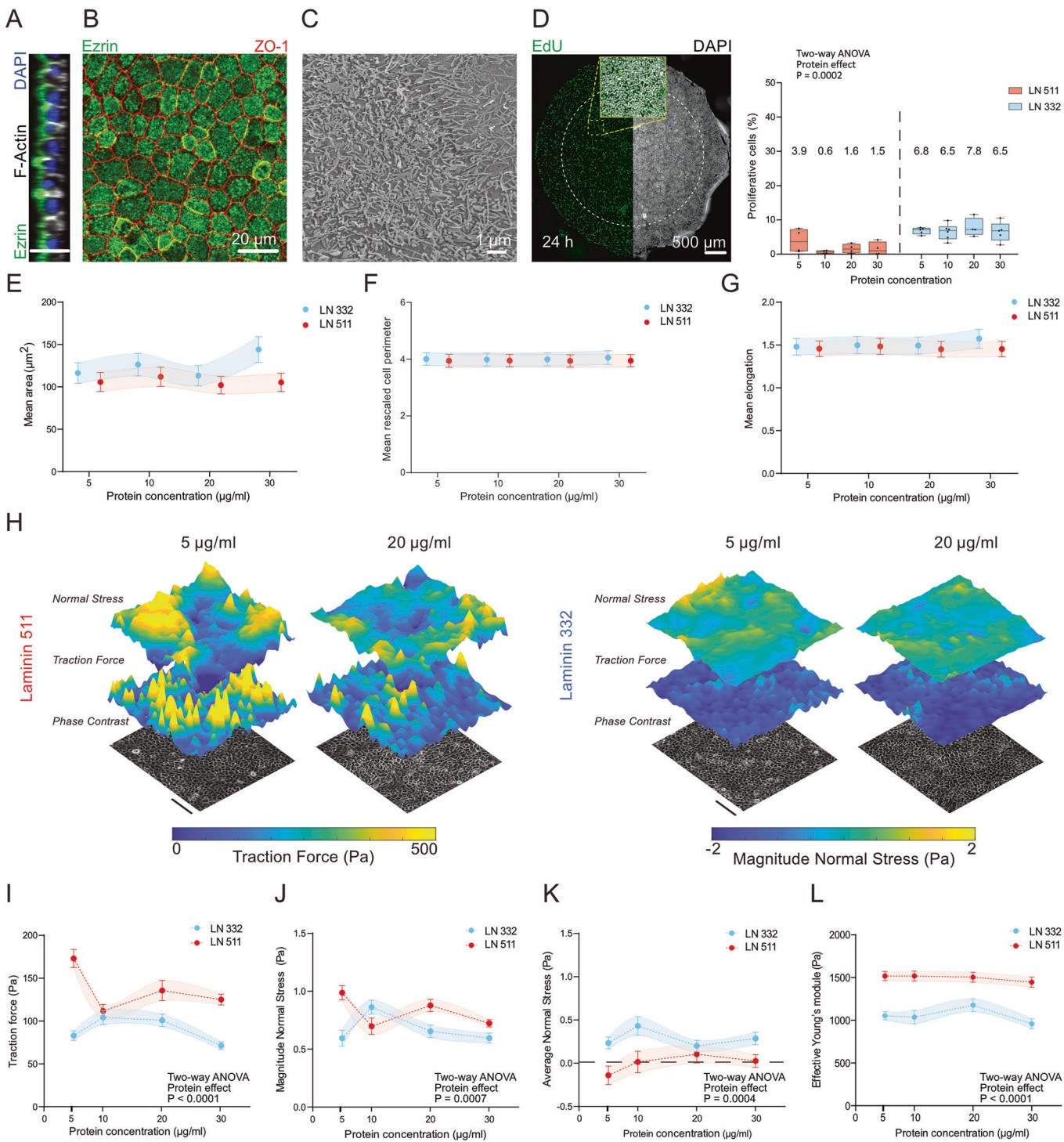

## Laminin 511 density influences contractility levels in RPE monolayers

The characterization of RPE cell topology and morphology in the retina suggested a relationship between laminin density in the basement membrane and monolayer mechanical status. To dissect the nature of this relationship, we established a bottom-up approach to examine the role of the specific laminin isoforms and their density

gradient. Human induced pluripotent stem cell-derived RPE (iRPE) cells have already been successfully employed to model mature RPE and to study their functionality (Song et al, 2023; Müller et al, 2018). Hence, we used iRPE cells to obtain monolayers on polyacrylamide (PAA) hydrogels with 4 kPa Young's modulus to best mimic homeostatic range of Bruch's membrane stiffnesses. This reductionist approach allowed us to decouple biochemical signals from mechanical cues and study the system under controlled conditions. iRPE cells

    

**Figure 2.  Reductionistic approach reveals the effect of laminin density on RPE biomechanics.**

iRPE monolayers obtained on PAA gel with Young's modulus of 4 kPa, coated with laminin 511 or laminin 332 in the presence of collagen type IV. After one week in culture, cells form a polarized monolayer as observed in a confocal cross-section stained for the microvilli marker Ezrin (A), mature tight junctions (ZO-1) (B), and distinct apical microvilli as observed in the scanning electron micrograph of the monolayer surface (C). DAPI staining marks the nuclei. (D) Cellular proliferation rate within 24 h obtained using EdU labeling assay for cells cultured on the PAA gel surface. A representative picture of the gel overview indicates the number of proliferating cells within one gel (left) compared to the total number of cells (right). The ratio of nuclei with incorporated EdU (per cent of proliferative cells) to the total number of nuclei labeled with DAPI was quantified by the fluorescent signal in the central region of the gel to avoid possible edge effects (white circle). The data show average values ± SEM from at least 4 independent experiments (biological replicates). Box and whisker plots display the median (center line), 25th–75th percentiles (bounds of the box), and minimum to maximum values (whiskers). All individual data points are shown. Statistical testing was performed using a mixed-effects model (REML) with the Geisser-Greenhouse correction, Tukey's multiple comparisons test. Morphometric analyses of the cells reveal the absence of significant difference in cellular mean area (E), mean rescaled perimeter (F) and mean elongation (G) between different laminin isoforms and densities from 3 independent experiments (biological replicates). (H) Representative traction force and monolayer stress profiles for different coating conditions and their quantifications (I–K) show a significant increase in contractility of iRPE monolayers at the lowest laminin 511 density. (L) Quantification of the effective Young's modulus of the apical side of monolayer in different conditions. Data are shown as the average of at least 6 different positions within the hydrogel (technical replicates) ± SEM from at least 4 independent experiments (biological replicates). Statistical analyses were performed using a mixed-effects model (REML) with the Geisser-Greenhouse correction, Tukey's multiple comparisons test. Source data are available online for this figure.

adhere and form a monolayer when seeded on PAA gels coated with different concentrations of laminin 511 (composed by laminin α5, β1, and γ1) and laminin 332, both in combination with collagen type IV (Fig. EV2A). The gels were functionalized according to the method described in Przybyla et al, (Przybyla et al, 2016) with specific concentrations of laminin, reaching saturation at a coating concentration of around 20 μg/ml (Fig. EV2B). The advantages of PAA gel as a culture substrate are its defined stiffness (Youngs's modulus of the gel) and the protein-repellent nature of the polymer (Funaki and Janmey, 2017), allowing control over the biochemical cues originating from the coating. In fact, the staining for fibronectin confirmed the absence of aberrant deposition of ECM proteins by the cells while cultured (Fig. EV2C). After a week of culture, iRPE cells reach a low proliferative state (less than 8% in 24 h) and form a columnar (junctional height ~10–15 μm) and polarized monolayer, as shown by the presence of apical microvilli (Figs. 2A–D and EV2D). Due to this reductionistic approach, certain coating conditions were not permissive to obtain mature and intact iRPE monolayers. For example, laminin 511 and collagen type IV alone or laminin 211 (composed by laminin α2, β1, and γ1) also in combination with collagen type IV (Fig. EV2A,E,F). In conclusion, this reductionistic approach suggests that mainly laminin 511 and 332, but not laminin 211, support RPE adhesion in vivo.

Next, following the logic of a comparative approach between in vivo and in vitro systems, we performed morphometric and topological analyses of the monolayers obtained on the different laminin isoforms and densities. In vivo, these analyses functioned as indicators of iRPE mechanical status. Nevertheless, in vitro, the quantification revealed a stable relationship among all laminin densities and isoforms, excluding their direct role in controlling monolayer morphology and topology (Figs. 2E–G and EV3A–D). However, we wondered if those laminin coatings may affect contractile forces of the monolayer. We used traction force and monolayer stress microscopy to quantify iRPE cell-matrix traction forces and cell-cell stresses in different conditions (Fig. 2H–K). The quantifications revealed an overall ability of laminin 511 to promote higher traction forces compared to laminin 332 (Fig. 2H,I), with the largest difference recorded at the lowest laminin density (170 Pa vs 80 Pa at 5 μg/ml) (Fig. 2I). Next, we used the information on the traction forces and monolayer height (Fig. EV2D) to calculate intercellular normal stresses in the different conditions as previously shown (Di Russo et al, 2021; Vishwakarma et al, 2018)

(Fig. 2J,K). Similarly, we observed a significant difference in the overall monolayer absolute stresses at the lowest coating density of laminins (5 μg/ml) with an average of ~1 Pa for laminin 511 and 0.6 Pa for laminin 332 (Fig. 2J). As expected, the stress vectors exhibit an average close to 0 Pa for both coatings and densities, indicating that monolayers are nearly in equilibrium despite the variations in absolute contractility (Fig. 2K). The difference in traction and stress at the lowest laminin 511 density hypothetically might be caused by a gradual increase in cell adhesion to collagen type IV, which remained constant across the conditions. We therefore tested the effect of collagen type IV on monolayer traction forces using laminin 332 as a control since laminin 511 alone does not support the formation of a monolayer (Fig. EV2E). The quantification shows an overall decrease (~40%) of traction in the presence of collagen type IV (Fig. EV3E), excluding the role of collagen type IV for the observed increased traction at the lowest laminin 511 density. Moreover, the clear distinction in traction between laminin 511 and laminin 332 strengthens the conclusion that the observed changes are exclusively laminin-induced.

Finally, we wanted to understand if the coating density could generally impact iRPE cell surface stiffness, which may regulate its apical interaction with the neural retina. Therefore, we performed nanoindentation of iRPE cells within the monolayers using a spherical probe of approximately 10 μm radius and quantified the average Effective Young's modulus of the monolayers (Fig. 2L). The data show that iRPE monolayers on laminin 511 are stiffer (~1.5 kPa) compared to laminin 332 (~1 kPa) regardless of the protein surface density.

Altogether, the biomechanical characterization of our model clearly indicates a divergent effect of the low density of laminin 511 versus laminin 332 on the overall RPE mechanical balance, promoting an increased contractility of the monolayer on laminin 511.

## Laminin-defined traction levels modulate RPE efficiency to phagocyte photoreceptor outer segments

Epithelial contractility levels have been shown to directly influence tissue functionality and reactivity (Vishwakarma et al, 2018; Di Russo et al, 2021; Park et al, 2015; Malinverno et al, 2017; Palamidessi et al, 2019; Balcioglu et al, 2020; Mongera et al, 2018; Pérez-González et al, 2021). Therefore, considering the difference

in iRPE mechanics on lower laminin density, we wondered if there is any direct relationship between the monolayer mechanical status and its functionality. To understand this, we tested the ability of RPE monolayers on PAA gels to perform their fundamental physiological function: to bind and phagocyte POS fragments, which were isolated from porcine eyes and fluorescently labeled with FITC (Fig. 3A–C). To quantitatively characterize the cells' ability to bind and internalize POS, we calculated phagocytosis efficiency as the ratio of internalized to total FITC-labeled fragments within the field of view. The quantification showed that, overall, at high laminin coating density (10–30 µg/ml) there is a comparable efficiency of phagocytosis between laminin 332 and 511, whereas a divergence is detected at the coating density of 5 µg/ml with reduced efficiency of iRPE on laminin 511 (~50%) (Fig. 3B,C). To determine whether this reduction in phagocytic efficiency was related to cellular maturity, we repeated the experiment on a four-week iRPE culture. The results demonstrated that the observed effect persisted over time, confirming that the differences were not influenced by culture duration (Fig. EV4A).

Furthermore, side-by-side comparisons showed no apparent differences in polarization markers, such as localization of Ezrin and ZO-1 in hand with E-cadherin and alpha-catenin, between laminin 511 and laminin 332, or between one- and four-week cultures (Fig. EV4B–F). These findings collectively indicate a fundamental difference between the two laminin isoforms in their ability to support RPE phagocytic activity.

These results highly correlate with the traction force levels in iRPE monolayers, suggesting a cause-effect relation between laminin-defined traction and RPE phagocytic activity. Hence, by orthogonal methods we addressed whether the efficiency of POS phagocytosis is directly affected by the level of RPE contractility independently of the biochemical nature of ECM adhesion (i.e., laminin isoforms or density) (Fig. 3D–J).

To systematically adjust the traction force, we decided to selectively tune either the ECM physical cues (i.e., hydrogel stiffness) or the biochemical nature of adhesion (i.e., laminins vs vitronectin) (Fig. 3D). We then assessed the iRPE efficiency in internalizing POS by comparing it with the maximum efficiency observed in iRPE on laminins (at 20 µg/ml). First, we used a stiffer substrate to test whether higher traction forces reduce iRPE phagocytic efficiency, as increasing stiffness from 4 to 18 kPa elevates traction forces without altering biochemical cues. We obtained iRPE monolayers on stiffer hydrogels (18 kPa) coated with laminin 511 or 332 (20 µg/ml) in the presence of collagen type IV (30 µg/ml). In these conditions, the iRPE developed a significantly increased traction force compared to the one on 4 kPa (~140 Pa for 511 and ~170 Pa for 332). This corresponded to a reduced efficiency of these monolayers for POS internalization (~40%) (Fig. 3E–G). Second, we measured the phagocytic activity and traction forces of monolayers obtained on 4 kPa hydrogels coated with vitronectin (250 µg/ml) to match the protein surface density of laminins. Also in this case, the analyses revealed a higher average traction force (~200 Pa), which corresponded to a reduced ability of iRPE to internalize POS (~36%) (Fig. 3E–G).

To distinguish the exclusive role of cellular contractility from the possible converging effects of ECM signaling, we inhibited or increased actomyosin contractility in iRPE monolayers using a ROCK inhibitor or a RhoA activator (Fig. 3H–J). The reduction of contractility was performed on iRPE monolayers cultured on soft

(4 kPa) surfaces coated with collagen type IV (30 µg/ml) with very low concentration of laminins (2.5 µg/ml), which showed per se a low activity for POS phagocytosis (~50%). This setup was designed to create conditions where laminin signaling has minimal influence, while phagocytic efficiency remains low. On the other hand, the increase of contractility was induced in iRPE on high-density laminins (20 µg/ml) in the presence of collagen type IV, where the highest phagocytic efficiency was observed. As expected, the drug treatments affected the ability of iRPE to develop traction forces with reduced traction after ROCK inhibitor treatment and increased after RhoA activation (Fig. 3H,J). Moreover, the quantification revealed a direct connection between actomyosin contractility and monolayer ability in POS internalization. Independently of the laminin type, iRPE phagocytic efficiency increased after ROCK inhibitor treatment (from ~50% to ~70%) and decreased after RhoA activator treatment (from ~70% to ~50%) (Fig. 3H,J).

Overall, these data demonstrate that low laminin 511 and not 332 density reduces RPE phagocytic efficiency by promoting high levels of traction forces.

## Integrin β4/β1 ratio and keratin network organization modulate RPE mechanical homeostasis along the visual axis

Epithelial cells in homeostasis mechanically couple their cytoskeleton to their basement membrane laminins via integrin receptors (Kozyrina et al, 2020; Di Russo et al, 2023). Intracellularly, integrin β1 subtypes interface with the actin cytoskeleton via talin, whereas integrin α6β4 binds via plectin to the keratin network (Alberts et al, 2017). Extracellularly, integrins β1 and β4 bind with a redundant number of proteins, including laminin α3 and α5 chains (Aumailley, 2012; Hynes, 2002). Therefore, we wondered if low densities of laminin 332 or 511 differently promote integrin-based adhesion in RPE, thus influencing traction forces and consequent functional differences. Hence, we performed a functional adhesion assay using blocking antibodies to address the engagement of specific integrin subtypes on laminin-coated hydrogels (Fig. 4A).

Blocking the different β chains (i.e., β1, β4, and [αv]β3) resulted in a significant reduction of adhesion for β1 (LN 511 ~ 62% and LN 332 ~ 75%) and no significant effect in the case of β4 and αvβ3 integrins. In the presence of integrin β4 blocking antibody, a tendency of reduction was observed on laminin 332. Hence, to avoid the effect of compensation by β1 integrin, we concurrently blocked integrin β1 and β4, achieving nearly complete inhibition of iRPE cell adhesion onto low-density laminin 332 (~94%) and a drop up to 80% on laminin 511 (Fig. 4A, left). A similar trend was also observed for integrin alpha chains, where a substantial reduction in adhesion occurred upon combining integrin α6- and α3-blocking antibodies, suggesting the predominant involvement of these two subtypes in the adhesion process (Fig. 4A, right).

Based on these findings, we stained target integrins on iRPE cells detached from mature monolayers and analyzed their relative surface presence during adhesion to the low-density laminin via flow cytometry (Figs. 4B–E and EV5). In line with the adhesion assay, the quantification revealed that cells adhering to the low-density laminin 511 present a significantly higher level of β1 integrin on their surface compared to cells adhering to laminin 332 (Figs. 4B,C and EV5E). Instead, the levels of β4, α3, and α6

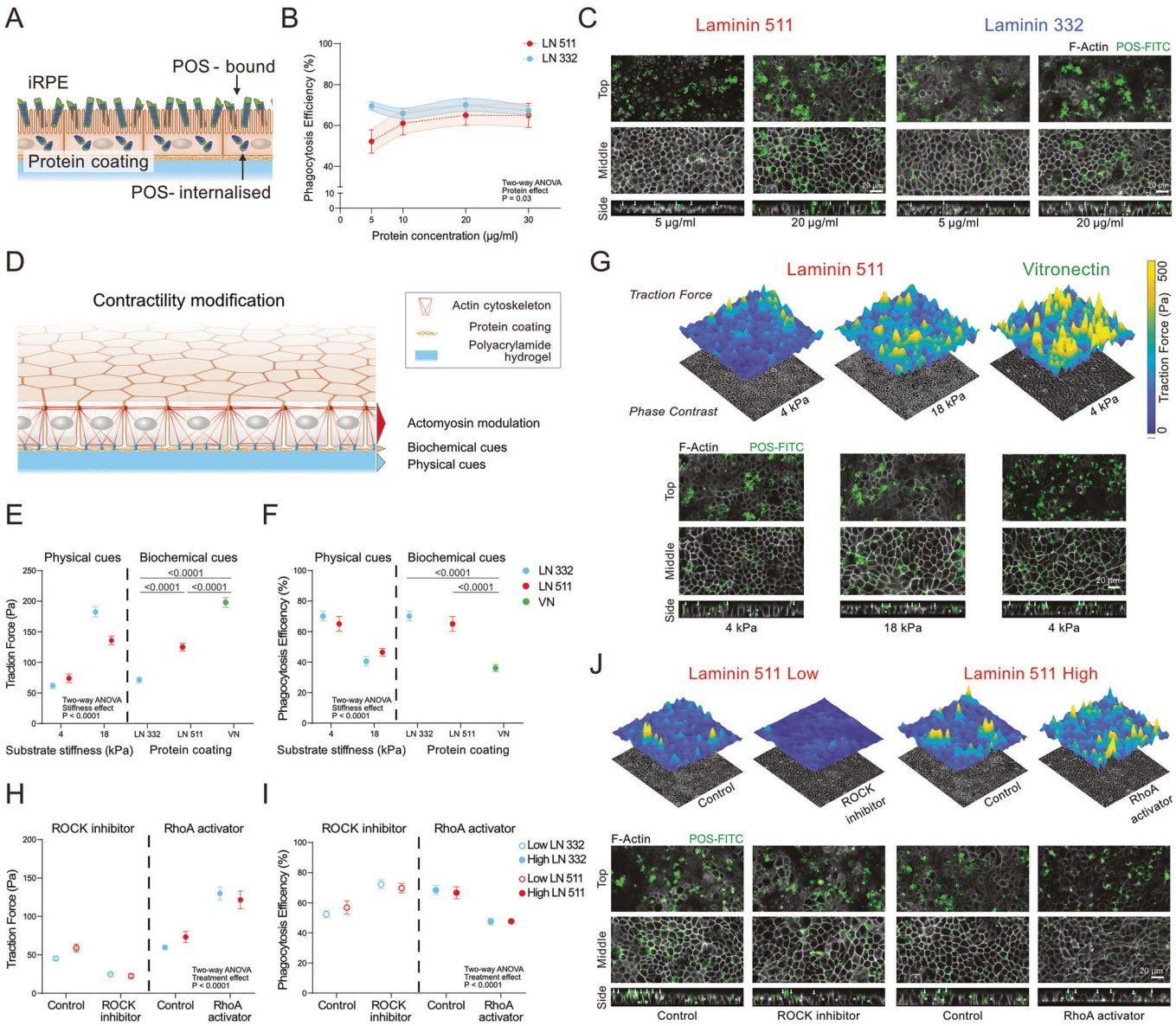

**Figure 3. Laminin-defined RPE contractility modulates POS phagocytosis.**

(A) Schematic representation of the photoreceptor outer segment internalization assay on iRPE monolayer. (B) Average phagocytosis efficiency from at least 3 different positions (technical replicates) ± SEM for different laminin isoforms and their coating conditions from at least 4 independent experiments (biological replicates). The phagocytosis efficiency is defined as the amount of internalized POS divided by total POS. Statistical significance was tested using a mixed-effects model (REML) with the Geisser-Greenhouse correction, Tukey's multiple comparisons test. (C) Representative confocal images of the iRPE cells on PAA hydrogel with laminin 511 or laminin 332 coating (in the presence of collagen type IV). The top row indicates FITC-labeled POS fragments at the apical surface of the cells. The middle row represents an optical section of the cell monolayer with POS fragments inside. The results of the POS phagocytosis assay are visible in the orthogonal projections (bottom row), where arrows and stars indicate bound and internalized POS, respectively. Note that the images presented in (C) and (J) are derived from the same experiment, and the top and middle laminin 511 20 µg/ml images are also presented in (J) as control of laminin 511-high condition. (D) Schematic representation of the experimental strategy to modulate monolayer contractility. Quantification of mean traction forces (from at least 5 technical replicates) (E) and phagocytosis efficiency (from at least 3 technical replicates) (F) ±SEM in iRPE upon different ECM physical (stiffness) and biochemical cues (coating) from at least 3 experiments (biological replicates). PAA gels were coated with laminin 511 (LN 511) or laminin 332 (LN 332) (in the presence of collagen type IV) or with vitronectin (VN). The data were tested using mixed-effects model (REML) with the Geisser-Greenhouse correction, Tukey's multiple comparisons test (physical cues) and Mann-Whitney test (biochemical cues); exact p-values are indicated on the respective comparisons. (G) Representative traction force plots and immunofluorescent images of POS internalization assay upon different physical or biochemical cues. Drug treatment affecting actomyosin contractility levels using ROCK inhibitor and RhoA activator shows inverse dependence between the level of monolayer contractility (traction forces) (H) and its ability to phagocyte POS fragments (I). Data are shown as mean from at least 3 technical replicates ± SEM from at least 3 experiments (biological replicates). The data were statistically analyzed using mixed-effects model (REML) with the Geisser-Greenhouse correction, Tukey's multiple comparisons paired test; exact p-values are shown on the respective comparisons. (J) Representative traction force plots and immunofluorescent images of POS internalization assay upon drug treatment. Note that the images presented in (C) and (J) are derived from the same experiment, and the top and middle of the control in laminin 511-high condition images are also presented in (C) for laminin 511 20 µg/ml condition. Source data are available online for this figure.

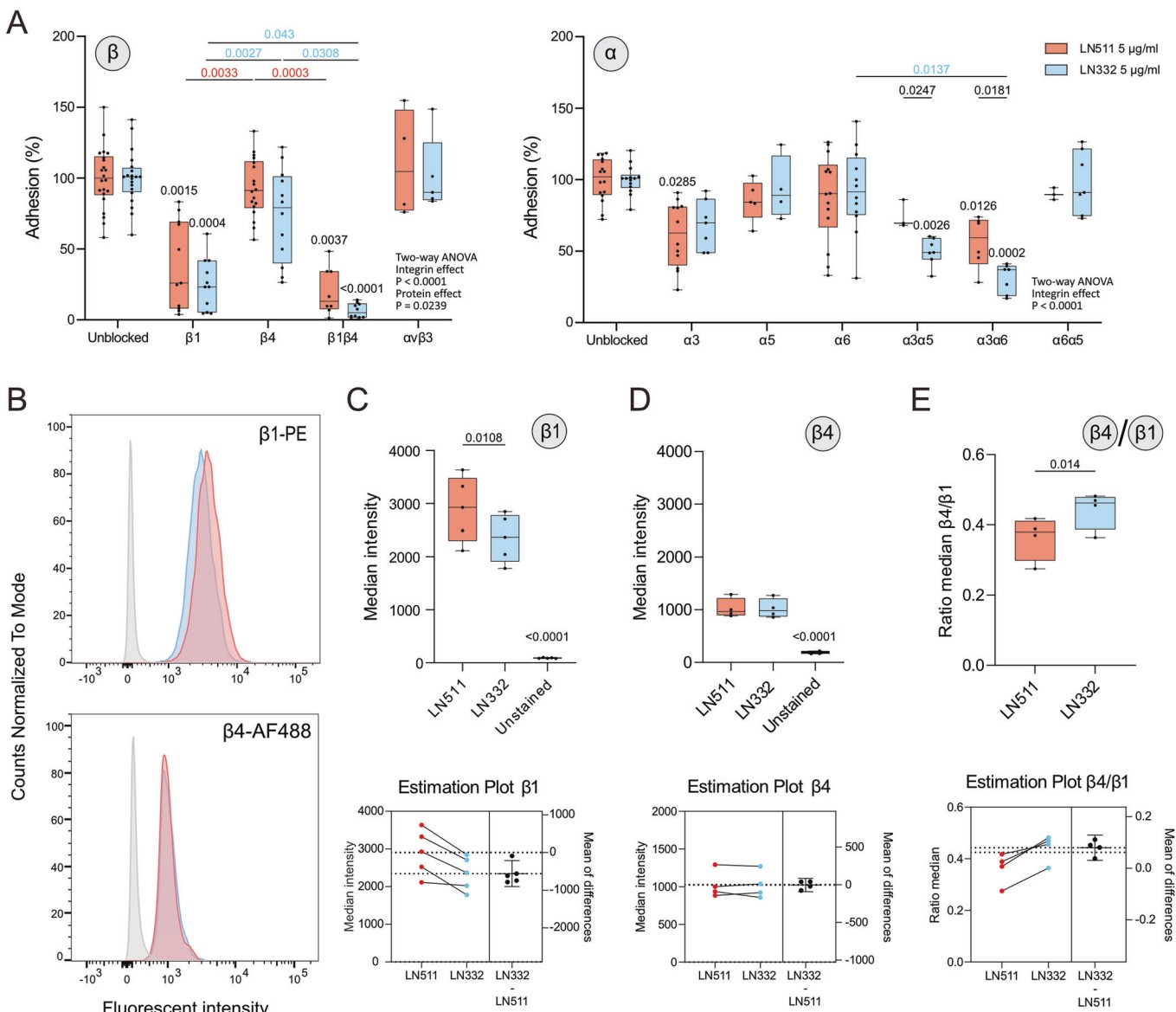

**Figure 4. Laminin density defines RPE mechanical balance via integrin β4/β1 ratio.**

(A) Box and whisker plots of iRPE cell adhesion percentage on low-density (5 µg/ml) laminin 332- or laminin 511-coated hydrogels after incubation with integrin β or α subunit-blocking antibodies and control (unblocked), based on at least 5 independent experiments (biological replicates), each with at least 3 technical replicates. Box and whisker plots display the median (center line), 25th–75th percentiles (bounds of the box), and minimum to maximum values (whiskers). All individual data points are shown. The data were statistically tested using a two-way ANOVA with Geisser-Greenhouse and Tukey correction. Exact p-values standing directly on the box and whisker show significance towards the control; p-values indicated with colors refer to the comparison between different integrins blocking (blue, red) or laminins (black). (B) Representative flow cytometry histograms of the surface integrin β1 (top) and β4 (bottom) expression of iRPE cells cultured for one week on low-density laminin 332 (light blue), laminin 511 (red), and unstained control (gray). (C–E) Box and whisker plots showing the distribution of intensity medians and the relative estimation plots from at least 4 independent experiments (biological replicates) for β1 and β4 integrins and their ratio. Box and whisker plots display the median (center line), 25th–75th percentiles (bounds of the box), and minimum to maximum values (whiskers). All individual data points are shown. Exact p-values are shown on the respective comparisons. Statistical significance was assessed using a paired t-test for comparison between laminins and one-way ANOVA with Dunnett's multiple comparison test for comparisons with unstained samples. Source data are available online for this figure.

integrins were comparable between low-density laminin 511 and 332 (Figs. 4D and EV5E–L). Furthermore, these results show that iRPE cells adhering to low-density laminin 332 present a significantly higher ratio of integrin β4 to β1 compared to cells adhering to low-density laminin 511 (Fig. 4E). Altogether, these data indicate the main role of integrins α6β1 and α3β1 to support

the adhesion to the low-density laminin 511 and integrins α6β1, α3β1, and α6β4 to low-density laminin 332.

The presence of the hemidesmosomal integrin α6β4 has been shown to reduce cellular traction forces (Wang et al, 2020) and sensitivity of nuclear mechanotransduction (Kechagia et al, 2023) via their role in nucleation and organization of the keratin

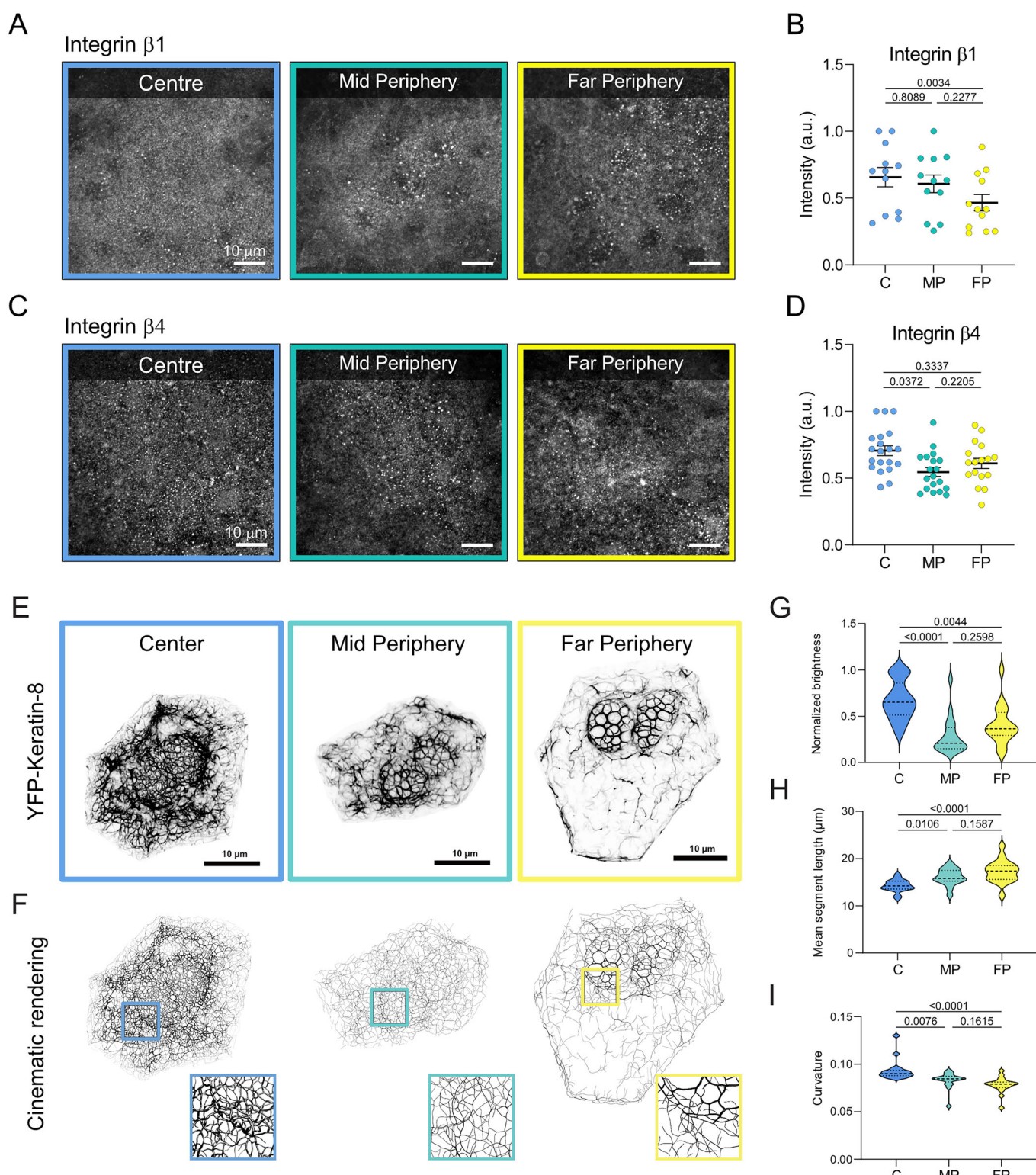

cytoskeleton (Moch and Leube, 2021; Kechagia et al, 2023). Encouraged by our in vitro findings, we were curious whether the different laminin densities in Bruch's membrane coincided with different amount of integrin β1 and β4 at the RPE-ECM interface and keratin network organization. Hence, we performed immunostaining of integrin β1 and β4 in in vivo (Fig. 5A,C). Although the data showed some variability, the results show that the amount of integrin β1 significantly decreases towards the periphery (Fig. 5A,B). In contrast, integrin β4 is rather similar between the central and peripheral regions of the retina (Fig. 5C,D).

◄ **Figure 5. Distribution of integrin β4 and β1 and keratin cytoskeleton organization along the visual axis in vivo.**

(A) Immunofluorescent staining of integrin β1 in RPE basal side at the centre, midperiphery, and far periphery of the retina. (B) Quantification of integrin β1 signal intensity in the specified regions. The data show average intensity values within the chosen region of interest (from at least four technical replicates) ± SEM from 3 independent mice (biological replicates). (C) Immunofluorescent staining of integrin β4 in RPE basal side at the centre, midperiphery, and far periphery of the retina. (D) Quantification of integrin β4 signal intensity in the specified regions. The data show average intensity values within the chosen region of interest (from at least four technical replicates) ± SEM from 3 independent mice (biological replicates). Scale bar 10 μm. (E, F) Representative images of RPE cells at different regions from YFP-keratin 8 mice and the resulting cinematic rendering of the keratin network. (G–I) Violin plots of normalized keratin brightness, mean segment length, and curvature in RPE from different retinal regions. The statistical significance was assessed using a one-way ANOVA. Data are from three independent mice (biological replicates). Exact p-values are indicated on the respective comparisons. Source data are available online for this figure.

Furthermore, using keratin 8-YFP knock-in mice (Schwarz et al, 2015), we imaged and segmented the keratin network in 3D using a recently developed workflow (Windoffer et al, 2022) (Fig. 5E–I). The analyses revealed that keratin network has thinner filaments (reduced brightness—Fig. 5G), higher segment length (Fig. 5H), and reduced segments curvature (Fig. 5I) from the centre to far periphery.

These findings align with our in vitro observations, where differential integrin expression is linked to the organization of the cytoskeletal network and traction forces.

Altogether, this evidence suggests the existence of a laminin-defined mechanical balance in RPE, which gradually shifts to a different keratin contribution in a more contractile monolayer toward the retinal periphery (Fig. 6).

## Discussion

Using a reductionistic approach, our findings demonstrate a direct role of laminin density in regulating RPE phagocytic efficiency through epithelial contractility. Furthermore, we propose that laminins play a crucial role in maintaining RPE mechanical homeostasis along the visual axis as suggested by the presence of a relative density gradient of specific laminin isoforms in the Bruch's membrane in vivo. The Bruch's membrane has the highest laminin content at the retinal centre, whereas at the far periphery, the laminin 332 and α5-containing isoforms have the lowest density. This laminin density gradient correlates with distinct topographical features and a gradual increase in RPE cell shape factor, where lower laminin density corresponds to more elongated cells. Cellular shape factor and cell-to-neighborhood relationship directly correspond to mechanical properties of the epithelial monolayers (Hannezo et al, 2014; Alt et al, 2017; Kaliman et al, 2021) and contractility (Park et al, 2015; Bi et al, 2015; Vishwakarma et al, 2018). Finally, the different arrangements of the keratin network within the RPE hint that, towards the periphery, the actin cytoskeleton may have a more prominent contribution to the epithelial mechanical status. Altogether, this evidence suggests higher epithelial contractility at the far periphery compared to the centre.

Due to technical limitations for the direct in vivo quantification of mechanical properties and contractility, we employed a human-relevant in vitro model using iRPE cells cultured on laminin-functionalized PAA gels. The stringent condition for ECM organization, availability and remodeling, created by the PAA gel,

revealed a direct role of low-density laminin 511 in increasing RPE contractility (higher traction forces and monolayer stress) and reducing the epithelium efficiency to phagocyte POS. Despite this increase in iRPE contractility, there were no observed alterations in cell shape or topology. This suggests that the monolayer topology and morphology in vivo may be defined by the convergence of several factors, such as RPE's anchorage to the *ora serrata* at the retinal periphery (Nobeschi et al, 2006). Moreover, our data rule out the exclusive role of laminins in defining RPE monolayer arrangement or stiffness. A direct implication of ECM organization in different areas of the retina cannot be excluded.

The effect of laminin α5 on cellular mechanics is not surprising. Laminin α5-containing isoforms have been shown to enhance cell stiffness and support better shear stress mechanotransduction (Di Russo et al, 2016) and intercellular tightness (Song et al, 2017) in the endothelium. As shown for the endothelium, RPE cells also preferentially adhere to this laminin by β1 integrins (i.e., α3β1 and α6β1 integrins). This is different from laminin 332, which additionally supports RPE adhesion via α6β4 integrins, i.e., hemidesmosomes. The role of hemidesmosomes in organizing the keratin network is crucial for the mechanical integrity of epithelial tissues (Hahn and Labouesse, 2001; Todorović et al, 2013) and was proposed to define their textile nature (Di Russo et al, 2023). More recently, data from different labs demonstrated that hemidesmosomes and the keratin network reduce cellular traction force (Wang et al, 2020) and protect the nucleus from mechanical strain (Kechagia et al, 2023). In agreement with the tensegrity model, cellular mechanics arise from a balance between cytoskeleton-driven forces, where contractile actin cytoskeleton is counter-balanced by intermediate filaments (Broussard et al, 2020; Ingber et al, 2014; Rölleke et al, 2023). Therefore, a higher integrin β4/β1 ratio in RPE adhering to low-density laminin 332, likely due to the more prevalent presence of the hemidesmosomes, could explain the lower traction forces measured under this condition, in contrast to low-density laminin 511, which promotes higher epithelial contractility. Consistently, our in vivo observations suggest that both β1 and β4 integrins decrease toward the retinal periphery. Moreover, a thinner and tenser keratin network in RPE in the far periphery and elongated nuclei suggest a higher contribution of the contractile actin cytoskeleton to tissue mechanics in this area. Nevertheless, technical challenges in quantitatively assessing the absolute amounts of ECM components, their receptors, and cellular contractility in vivo leave some open questions. These include whether the relative abundance of laminin isoforms varies along the visual axis and whether the ratio of β4 integrins (hemidesmosomes)

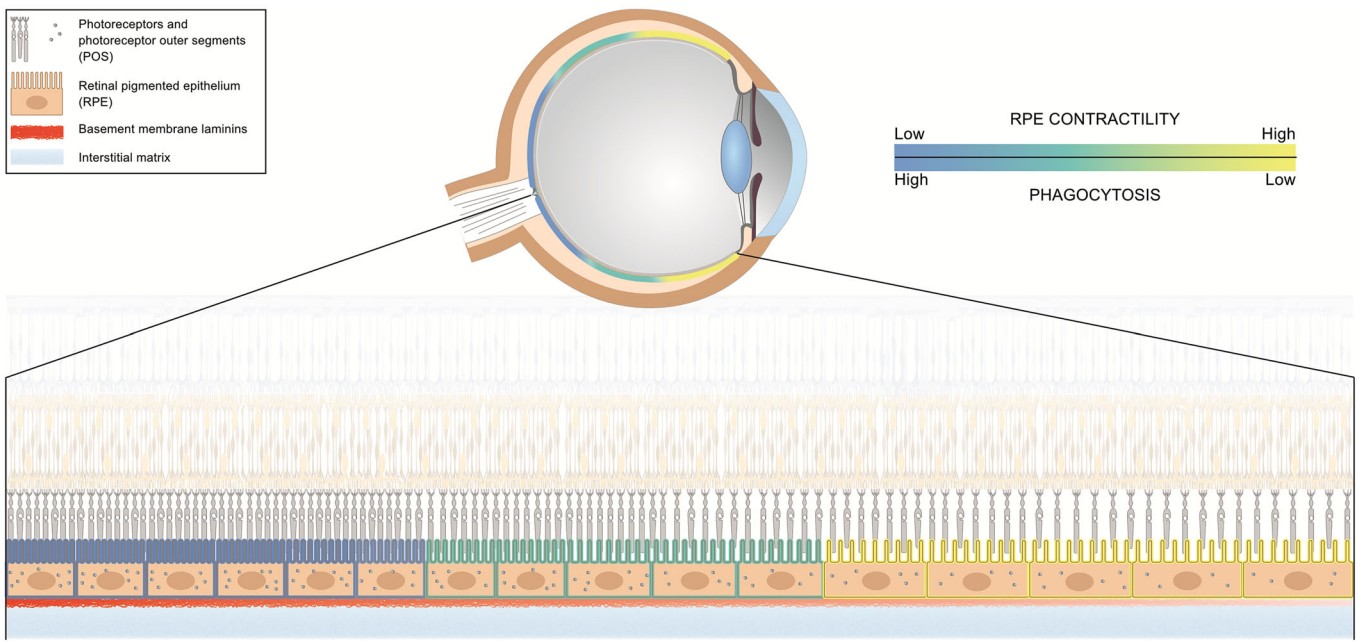

**Figure 6. RPE mechanical homeostasis along the visual axis.**

Schematics of ECM-orchestrated differences in the retinal tissue shows how high basement membrane laminin density in the macula ensures an optimal mechanical status that supports high phagocytic activity. Laminin isoforms 511 and 332, but not laminin 211, play a pivotal role in this regulation. Low laminin density in the retinal periphery increases RPE contractility, which corresponds to reduced phagocytosis.

to β1 integrins decreases with distance from the optic nerve, potentially increasing mechanical strain on the RPE, as suggested by morphometric analyses.

The fact that ECM density reduction increases epithelial traction has been previously proposed based on the reduction of integrin β1 receptor engagement in vitro (Oria et al, 2017; Di Russo et al, 2021). Here, for the first time, we show the physiological relevance of this density and the importance of the β4/β1 ratio for epithelial mechanics. Future studies will need to focus on the molecular mechanisms regulating epithelial mechanical homeostasis by laminin receptors' density. This is particularly relevant for retinal aging, given the reported reduction in hemidesmosomal attachment sites in RPE (Anderson et al, 1995).

A study on stem cell-derived RPE has previously shown that the organization of the actin cytoskeleton can predict the ability of differentiated cells to phagocyte POS (Müller et al, 2018). Particularly, the presence of F-actin stress fibers impaired the phagocytic activity of these cells. These data and our findings strongly suggest that high contractility in RPE is detrimental to its homeostasis and function. RPE apical microvilli interact with POS mostly via the Mer tyrosine kinase receptor (MerTK) and the integrin αvβ5 (Kwon and Freeman, 2020). The activation of MerTK is thought to be involved in the inside-out activation of integrin αvβ5 with the recruitment of talin, vinculin and interaction with F-actin (Kwon and Freeman, 2020). These focal adhesion-like structures have already been shown to mediate phagocytosis in macrophages and to be mechanosensitive (Jaumouillé et al, 2019). Hence, the formation of this apical molecular clutch may be sensitive to the overall cellular contractility defined by the ECM

substrate and explains the lower efficiency of POS internalization observed in this work. Furthermore, the homophilic interaction between POS and RPE via the neural cell adhesion molecule (N-CAM) may also be impacted by increased RPE contractility affecting the bond lifetime, thus, binding efficiency. It is important to note that our evaluation of phagocytic efficiency in vitro assumes an equal degradation rate of internalized POS across conditions. This assumption is based on previous findings showing that iRPE primarily degrades POS within a timeframe of six to 24 h rather than within the four-hour POS exposure period used in our assay (Almedawar et al, 2020; Schreiter et al, 2020). Nonetheless, the contribution of the degradation rate to the results cannot be entirely ruled out under our culture conditions. Future studies should focus on better understanding the dynamics of this process in relation to RPE contractility, including the rates of phagosome maturation and degradation.

Overall, our work underlines the important role of RPE mechanobiology for outer retina physiology and opens important questions about its role in retinal aging. ECM remodeling during aging may affect cell adhesion and shift RPE mechanical homeostasis out of its optimal range, making it more prone to phenotypical changes. Overcoming this threshold may cause sight-impairing diseases such as (high) myopia and age-related macular degeneration. Moreover, the use of Rho kinase inhibitors, as in glaucoma treatment (Al-Humimat et al, 2021; Mandell et al, 2011), may be implemented to prevent the progression of RPE-specific diseases. Undoubtedly, further studies will be necessary to elucidate to what extent RPE contractility is involved in outer retina diseases.

　　　　　　　　　　　　　　　　　　　　　　　

# Methods

### Reagents and tools table

| Reagent/Resource | Reference or Source | Identifier or Catalog Number |
| --- | --- | --- |
| **Experimental models** | | |
| BALB/c Mice | University Hospital RWTH Aachen | https://www.jax.org/strain/000651 |
| Keratin 8-YFP C57BL/6 Mice | Schwarz et al, 2015 | N/A |
| hiPSC-derived RPE | FUJIFILM Cellular Dynamics | Cat #R1102 |
| **Antibodies** | | |
| Rabbit anti-mouse laminin 111 (pan-Laminin), rabbit serum (1:2000) | Gift from Lydia Sorokin | N/A |
| Rat anti-mouse laminin α1 (clone 200), rat hybridoma supernatant | Gift from Lydia Sorokin | N/A |
| Rat anti-mouse laminin α2 (clone 4H8-2), rat hybridoma supernatant | Gift from Lydia Sorokin | N/A |
| Rabbit anti-human laminin 332, rabbit serum (1:2000) | Gift from Monique Aumailley | N/A |
| Rabbit anti-mouse laminin α4, rabbit serum (1:1000) | Gift from Lydia Sorokin | N/A |
| Rabbit, anti-mouse laminin α5, rabbit serum (1:1000) | Gift from Lydia Sorokin | N/A |
| Rabbit anti-mouse collagen type IV (12.5 µg/ml) | Merck Millipore | Cat #AB756P |
| Rabbit anti-mouse collagen type I, (5 µg/ml) | Novusbio | Cat #NB600-408 |
| Rabbit anti-mouse elastin (5 µg/ml) | Abcam | Cat #ab21610 |
| Phalloidin iFluor 647 (1:500) | Abcam | Cat #ab176753 |
| Phalloidin iFluor 488 (1:500) | Abcam | Cat #ab176759 |
| Rabbit anti-human ZO-1 | Thermo Fisher | Cat #61-7300 |
| Mouse anti-human ezrin (clone 3C12) | Abcam | Cat #ab4069 |
| Rabbit anti-human fibronectin | Sigma | Cat #F3648 |
| Mouse anti-human e-cadherin | BD Transduction lab | Cat #610182 |
| Rat anti-α-18 α-catenin | Gift from Akira Nagafuchi | N/A |
| Rabbit anti-human/mouse integrin β4 | Abcam | Cat #ab236251 |
| Rabbit anti-mouse integrin β1 | Gift from Staffan Johansson | N/A |
| **Antibodies used for adhesion assay** | | |
| Mouse anti-integrin β1 (clone P5D2) (1:100) | Santa Cruz | Cat #sc-13590 L |
| Mouse anti-integrin β4 (clone ASC-8) (1:50) | Merk | Cat #MAB2059Z |
| Mouse anti-integrin αvβ3 (clone LM609) (1:50) | Merk | Cat #MAB1976Z |
| Mouse anti-integrin α3 (clone P1B5) (1:50) | Merk | Cat #MAB1952Z |
| Mouse anti-integrin α5 (clone P1D6) (1:50) | Merk | Cat #MAB1956Z |
| Rat anti-integrin α6 (clone GOH3) (1:100) | Santa Cruz | Cat #sc-19622 L |
| **Antibodies used for flow cytometry** | | |
| Mouse anti-integrin β1-PE (clone 12G10) (1 mg per million cells) | Santa Cruz | Cat #sc-59827 PE |
| Mouse anti-integrin β4-Alexa Flour 488 (clone A9) (1 mg per million cells) | Santa Cruz | Cat #sc-13543 AF488 |
| Mouse anti-integrin α3-Alexa Flour 647 (clone P1B5) (1 mg per million cells) | Santa Cruz | Cat #sc-13545 AF647 |
| Rat anti-integrin α6-Brilliant Violet 421 (clone GOH3) (5 µl per million cells) | BioLegend | Cat #313623 |
| **Secondary antibodies** | | |
| Goat anti-mouse-IgG AF488 (4 µg/ml) | Invitrogen | Cat #A11001 |
| Goat anti-mouse-IgG AF594 (4 µg/ml) | Invitrogen | Cat #A11005 |
| Goat anti-mouse-IgG AF647 (10 µg/ml) | Jackson/Dianova | Cat #111-605-144 |
| Goat anti-rabbit-IgG AF594 (4 µg/ml) | Invitrogen | Cat #A11012 |
| Goat anti-rabbit-IgG AF647 (10 µg/ml) | Molecular Probes | Cat #A21235 |
| Goat anti-rat-IgG AF555 (4 µg/ml) | Invitrogen | Cat #A21434 |
| Goat anti-rat-IgG AF647 (10 µg/ml) | Thermo Fisher | Cat #A21247 |
| **Chemicals, Enzymes and other reagents** | | |
| Accutase Solution | Sigma-Aldrich | Cat #A6964-100ML |
| Albumin bovine (BSA) Fraction V | Serva | Cat #11930.03 |
| CellAdhere Dilution Buffer | StemCell Technologies, 07183 | Cat #07183 |
| CELLBANKER 2® freezing media | Amsbio LLC | Cat #11891 |
| Dimethyl sulfoxide (DMSO) | Sigma-Aldrich | Cat #D4540 |
| Dulbecco's Phosphate-Buffered Saline (DPBS) Calcium/Magnesium | Thermo Fisher Scientific | Cat #14040-091 |
| Ethylenediaminetetraacetic acid (EDTA) | Sigma-Aldrich | Cat #E5134-1KG |
| EDTA-free Protease Inhibitor Cocktail | Roche | Cat #4693132001 |
| Gentamicin | Gibco | Cat #15750-060 |
| Hoechst 33342 | Thermo-Fischer Scientific | Cat #62249 |
| Human protein S | Coachrom Diagnostica | Cat #pp012A |
| Hydrocortisone Solution | Sigma-Aldrich | Cat #H6909-10ML |

| Reagent/Resource | Reference or Source | Identifier or Catalog Number |
|---|---|---|
| KnockOut Serum | Thermo Fisher Scientific | Cat #10828-028 |
| MEM α, GlutaMAX™ Supplement, no nucleosides | Gibco | Cat #32561-029 |
| N-2 supplement | Gibco | Cat #17502048 |
| Recombinant Human MFG-E8 Protein, CF | R&D systems | Cat #2767-MF-050 |
| Paraformaldehyde, prilled, 95% | Sigma-Aldrich | Cat #441244-1KG |
| Taurine | Sigma-Aldrich | Cat #T0625 |
| TrypLE Express dissociation agent | Gibco | Cat #12605-010 |
| Trypsin-EDTA solution | Sigma-Aldrich | Cat #T4049-500ML |
| **Inhibitors** | | |
| RHO ACTIVATOR II | Cytoskeleton | Cat #CN03-A |
| ROCK-Inhibitor (Y-27632) | Sigma-Aldrich | Cat #0503-1MG |
| **Extracellular matrix proteins** | | |
| Biolaminin 211 | BioLamina | Cat #LN211-02 |
| Biolaminin 332 | BioLamina | Cat #LN332-0502 |
| Biolaminin 511 | BioLamina | Cat #LN511-0502 |
| Human Collagen Type IV, Collagen from human placenta | Sigma-Aldrich | Cat #C7521-10MG |
| Vitronectin XF | StemCell Technologies | Cat #07180 |
| **Software** | | |
| GraphPad Prism 10.4 | https://www.graphpad.com | |
| ImageJ | https://imagej.nih.gov/ij/index.html | |
| DataViewer V2.4.0 | https://www.optics11life.com | |
| FlowJo | https://www.flowjo.com | |
| Cellpose | https://www.cellpose.org | |
| MATLAB | https://www.mathworks.com/products/matlab.html | |
| Affinity Designer | https://affinity.serif.com | |
| **Other** | | |
| Confocal LSM 710 | Carl ZEISS AG | |
| Axio Observer 7 Inverted Microscope | Carl ZEISS AG | |
| ApoTome.2 Fluorescence Microscope | Carl ZEISS AG | |
| Chiaro Nanoindenter | Optics11 Life | |
| LSRFortessa Cell Analyser | BD Biosciences | |
| Dimension Icon Atomic Force Microscope | Bruker | |

## Retina flat mount preparation

Retina flat mounts were prepared from BALB/c mice of 25–30 weeks of age, either female or male. After sacrifice, the eyes were enucleated and fixed with 2% (wt/vol) paraformaldehyde (PFA) for 4 h at 4 °C rotating. To best visualize the RPE layer, the eyes were prepared as flat mounts according to published protocol (Claybon and Bishop, 2011). Briefly, the cornea, the iris and the lens were removed, and the eyecup was cut into eight longitudinal slices to allow its spreading. Afterwards, the neurosensory retina was gently detached exposing the RPE layer adhering to its ECM for fluorescent staining and imaging. The animal care and sacrifice were conducted according to the German Animal Welfare Act (LANUV, reference number 84- 02.04.2015.A190).

## Atomic force microscopy of Bruch's membrane

Upon enucleation, murine eyes were freshly dissected using cold phosphate-buffered saline without $Ca^{2+}$ and $Mg^{2+}$ ($PBS^{-/-}$) and containing protease inhibitors as instructed by the manufacturer (Roche, 4693132001). Dissected eyes were denuded by the RPE layer via incubation of the tissue for 60 min with 50 mM Tris–HCl buffer plus 20 mM ethylenediamine tetraacetic acid (EDTA), pH 7.4, followed by 20 min incubation in 50 mM Tris–HCl buffer and two washes of 10 min with 1 M NaCl, pH 7.4. Each solution contained freshly added cOmplete™, EDTA-free Protease Inhibitor Cocktail (Roche, 4693132001). Next, the prepared eyes were glued to a surface using Corning Cell-Tak Cell and Tissue Adhesive (Corning, 354240). Cantilevers with a spherical colloidal tip of 6.62 μm diameter (Nano and More GmbH, CP-PNPS-SiO-C-5) were used to indent 200 nm into the decellularised Bruch's membrane in different regions of the eye. The experiments were performed in a Dimension Icon atomic force microscope (Bruker Co. USA). Probes were calibrated using the thermal oscillation method, built into the software (Nanoscope 9.4). Force distance curves were recorded with an approach and retraction speed of 1 μm/s, with a relative maximal force set typically to 2 nN. The maximal range of the Z distance was set to 2.5 μm. All curves were exported to ASCII files with the analysis software of the instrument. The approach part of the curves was analyzed using a custom-written script in Python (https://www.python.org) using the Hertz model (Eq. (1)). The Poisson ratio was assumed to be $\nu = 0.3$. The last 40–50% of the data points are fit to a line and subtracted as background.

$$F(\delta) = \frac{4}{3} \frac{E}{1-\nu^2} \sqrt{R}(\delta - \delta_0)^{3/2} \tag{1}$$

The model fits the whole curve using a nonlinear fit (leastsq) from Scipy (https://www.scipy.org) finding both Young's modulus and the contact point (code is available under: https://github.com/tomio13/AFMforce).

## Immunofluorescence of retinal tissue

Fluorescent staining of retina flat mounts was performed as follows. After fixation and dissection, the tissue was incubated for 1 h in 1% bovine serum albumin (BSA) (wt/vol) in $PBS^{-/-}$ containing 0.1%

(vol/vol) Triton X-100, followed by overnight staining at 4 °C with primary antibodies or phalloidin dissolved in PBS$^{-/-}$ solution of 1% (wt/vol) BSA and 0.1% (vol/vol) Triton X-100. Secondary antibodies were diluted in PBS$^{-/-}$-containing 0.1% (vol/vol) Triton X-100 and incubated with the tissue at room temperature for 2 h. Finally, the retinas were incubated for 15 min with 5 µg/ml Hoechst 33342 (Thermo-Fischer Scientific, 62249) in PBS$^{-/-}$ before being mounted on glass slides with Mowiol mounting medium (Sigma-Aldrich). Primary and secondary antibodies used in this work are listed in Reagents and Tools Table.

## Retinal extracellular matrix quantification

To quantify the relative amount of ECM protein in the Bruch's membrane upon immunofluorescence staining, Z-stacks (0.5 µm optical slices) from the different regions were obtained using a Zeiss LSM 710 (Carl Zeiss) confocal microscope with a Plan-Apochromat 63 x/1.4-NA, DIC M27 oil immersion objective using unified illumination settings. The distance from the optic nerve was monitored using the automated stage of the microscope and setting the optic nerve as an origin. Within the eye, Z-stacks for each region were recorded per eight longitudinal slices, respectively. Mean fluorescent intensity in single optical sections identified as BrM was measured using Fiji software (National Institutes of Health, USA). Manually defined regions of interest (ROIs) were selected to avoid artefacts due to tissue tearing during preparation (Fig. 1I "stars", Fig. EV1N). Three to five ROIs were measured for each stack, and the fluorescence intensity of all data was normalized to the highest measured value in the eye.

## Keratin-8 network segmentation and characterization

Retinas for keratin-8 analysis were prepared as flat mounts from homozygous keratin 8-YFP knock-in mice (Schwarz et al, 2015). In each region (centre, mid periphery and far periphery), z-stacks of six cells were recorded per animal using a Plan-Apochromat 63 x/1.4-NA, DIC M27 oil immersion objective in Airyscan mode at a Zeiss LSM 710 (Carl Zeiss), for a total of 54 analyzed cells. To ensure the analysis of only individual cells, the signal within individual cells was cropped from confluent monolayers using standard tools in Fiji. Cell borders were manually defined for each z-slice and the cell exterior was set to a gray value of 0. The obtained images were segmented and processed to obtain a keratin network model for each cell as previously described (Windoffer et al, 2022). Finally, custom-made Matlab scripts were used to analyze the network properties including brightness, segment length, and curvature (Windoffer et al, 2022).

## iRPE culture

Differentiated iRPE were purchased cryopreserved from FUJIFILM Cellular Dynamics and after thawing, seeded in 24-well plates coated with 2.5 µg/ml Vitronectin XF (StemCell Technologies, 07180) to acclimatise the cells to culture condition and increase the cell number before use at a density of 1600–1800 cells/mm². Cells were cultured to confluence in a humidified incubator at 37 °C and 5% CO$_2$ and then, after detachment, reconstituted in CELLBAN-KER 2® freezing media (Amsbio LLC, 11891) and kept in liquid nitrogen for future use. Cell detachment for cryopreservation or

experiments was achieved through 30 min of incubation with 5 mM EDTA in PBS without Ca$^{+2}$ and Mg$^{+2}$ (PBS$^{-/-}$) followed by TrypLE Express dissociation agent (Gibco, 12605-010) for 5 min at 37 °C. Prior to seeding, cells were centrifuged at 300 g for 5 min and resuspended in the media at the desired density. For the experiments, cells were thawed again and cultured for up to two weeks until confluence before use. For seeding on polyacrylamide hydrogels, cells were seeded at a density of 5000–6000 cells/mm², corresponding to the RPE density in the human macula (Bhatia et al, 2016) and cultured for one week. Cells were cultured in iRPE medium composed of MEM α (Thermo Fisher, 12549-089), 5% (vol/vol) knock-out serum replacement (Gibco, 10828-028), 1% (vol/vol) N-2 supplement (Gibco, 17502-048), 55 nM hydrocorti-sone (Sigma-Aldrich, H6909), 250 µg/ml taurine (Sigma, T0625), 14 pg/ml triiodo-L-thyronine (Sigma, T5516) and 25 µg/ml genta-micin (Gibco, 15750-060). The medium was exchanged every second day. Cells were tested for mycoplasma contamination on regular basis.

## Hydrogel preparation and functionalization

To obtain flat hydrogel substrates, a mixture of 40% acrylamide (Bio-Rad, 1610140) and 2% bis-acrylamide (Bio-Rad, 1610142) in distilled water was first prepared. The following ratios of acrylamide/bis-acrylamide in the mixture were used to reach different hydrogel Young's moduli quantified by nanoindentation: 5%/0.1% (wt/vol) for 4 kPa and 10%/0.07% (wt/vol) for 18 kPa. Then, the mixture was combined with 0.5% of (vol/vol) ammonium persulfate (Sigma-Aldrich, 248614, 10% (wt/vol) stock solution in distilled water) and 0.05% (vol/vol) N'-tetramethylethylenediamine (TEMED) (Sigma-Aldrich, 411019), and pipetted between glutaraldehyde-activated glass bottom petri dishes (Cellvis, D35-20-0-N) and a hydrophobic glass coverslip. If hydrogels were used for traction force microscopy, 1 µm yellow-green carboxylate-modified polystyrene microbeads (Sigma, L4655) were included in the solution.

ECM proteins were crosslinked on the hydrogels' surface according to the protocol described by Przybyla L. and colleagues (Przybyla et al, 2016). Briefly, for functionalization solution, we prepared stocks of 0.5 M HEPES NaOH pH6 and 0.2% (wt/vol) bis-acrylamide in distilled water, 0.2% (vol/vol) tetramethacrylate (Sigma, 408360) and 3% (wt/vol) hydroxy-4'-(2-hydroxyethoxy)-2-methylpropiophenone (Sigma, 410896) in ethanol. Acrylic acid N-hydroxysuccinimide ester (Sigma, A8060) was initially dissolved in DMSO (10 mg/ml) and later reconstituted in 50% ethanol for the final concentration of 0.03% (wt/vol). Calculating 0.5 ml solution per gel, we combined 216.65 µl of sterile water, 50 µl of 0.5 M HEPES NaOH pH 6, 32.5 µl of ethanol and 25 µl of bis-acrylamide and degassed under vacuum for 20 min. Then, 5 µl tetramethacry-late, 4.165 µl hydroxy-4'-(2-hydroxyethoxy)-2-methylpropiophe-none and 5 µl of the ester reconstituted in 161.65 µl of 50% ethanol were added to the degassed solution. The completed functionalization solution was applied over the surface of hydro-gels, previously partly dehydrated in a 70% ethanol solution for 5 min, and the hydrogels were exposed for 10 min to ultraviolet light. After sequential double washing with 25 mM HEPES buffer and PBS$^{-/-}$ for a total of 20 min, the gel surface was incubated overnight at 4 °C with the desired ECM protein solution. Human recombinant laminin-511 (Biolamina, LN511-0202) and laminin-

332 (Biolamina, LN332-0202) were prepared at the desired concentration in PBS containing $Ca^{2+}$ and $Mg^{2+}$ ($PBS^{+/+}$) mixed with 30 µg/ml collagen type IV (Sigma-Aldrich, M7027-100G). Vitronectin (StemCell Technologies, 07180) was used without dilution at 250 µg/ml. Finally, the obtained hydrogels were washed in $PBS^{+/+}$ and sterilized by 30 min ultraviolet light irradiation before their application as cell culture substrates.

To represent surface protein saturation, gels were chemically crosslinked and coated with laminin-511 at concentrations of 5, 10, 20, or 30 µg/ml along with collagen type IV (30 µg/ml) in $PBS^{+/+}$. After overnight incubation at 4 °C, gels were thoroughly washed with $PBS^{+/+}$ to remove unbound protein, fixed with 2% PFA for 10 min at room temperature, blocked with 1% BSA and stained with primary Anti-mouse laminin α5 (gift from Lydia Sorokin) and secondary Anti-mouse IgG AF594 (Invitrogen, A11005). Between all steps, gels were washed with $PBS^{-/-}$ for at least 1 h. Stained gels were imaged with a 63x objective on ApoTome.2 Fluorescence Microscope (Carl Zeiss), with at least 10 regions per gel. The average intensity of each image was measured using Fiji software, and the final results were plotted in GraphPad Prism 10.

## Immunofluorescence staining of iRPE

Cells on hydrogels were first quickly washed with warm (~37 °C) $PBS^{+/+}$ then fixed with warm (~37 °C) 2% (wt/vol) PFA for 8 min at room temperature. After fixation, cells were permeabilized with 0.3% (vol/vol) Triton X-100 for 2 min and blocked with 1% (wt/vol) BSA for 30 min at room temperature. Primary antibodies or Phalloidin-iFluor 647 Reagent (Abcam, ab176759) in PBS containing 1% (wt/vol) BSA were incubated overnight at 4 °C. The following day, secondary antibodies were diluted in a PBS solution containing DAPI (1 µg/ml, Invitrogen D1306) and incubated with the sample for 30 min at room temperature. Primary and secondary antibodies used in this work are listed in Reagents and Tools Table. The samples were imaged in PBS solution to match the immersion type of the objective. Cells were imaged using C-Apochromat 40x/1.2 W Corr M27 water immersion objective on Zeiss LSM 710 confocal microscope with Airyscan capability (Carl Zeiss).

## Live staining

For live imaging of F-actin, iRPE cells cultured on PAA gels were incubated with 500 nM of a fluorogenic F-actin probe SirActin (Spirochrome, SC001) containing medium for at least 2 h before imaging. The samples were imaged using C-Apochromat 40x/1.2 W Corr M27 water immersion objective on Zeiss LSM 710 confocal microscope with Airyscan capability (Carl Zeiss). Orthogonal views from Z-stacks acquired via confocal live-imaging of F-actin (SirActin) were used to measure the average junctional height of cultured iRPE monolayers using Fiji image processing package (National Institutes of Health, USA).

## Morphometric and topology analyses

Microscopy images of RPE in vivo and in vitro were segmented using the cellpose software (Stringer et al, 2021) (www.cellpose.org) applied on F-Actin micrographs. Segmented cells were imported into Fiji as ROIs and used to generate skeletonized images. The skeletonized images were then imported into custom-written MATLAB software (Kaliman et al, 2016). Prior to analysis, the software removes 15 pixels on the sides of the images to remove poor mask definition by cellpose (Fig. EV1B), and then applies a Gaussian filter with a standard deviation of 2 pixels (function "imgaussfilt") to correct the pixel connectivity on the masks provided by cellpose. Finally, the software extracts information about the cell morphology (cell area, perimeter, and elongation) and topology (cell neighbors) as previously described (Kaliman et al, 2016). For these analyses, the cells at the edge of the picture were not considered to avoid any segmentation error (Fig. EV1C). For in vivo images, in total, 24 images from 3 different mice have been analyzed for the "centre" and "mid periphery" regions of the retina (8 images each) and 23 images for the "far periphery" region (8-8-7 images for the 3 mice, respectively). This corresponds to a total cell number of 5575 cells for the center, 7957 for the mid periphery and 7863 for the far periphery. For in vitro images, 71 images were analyzed (at least 2 images per experiment). Each image provided in between 86 and 312 cells. Because of segmentation error, to remove unnaturally large cells, cells with a rescaled area (see below) higher than 2, and cells with a neighborhood larger than 10 cells have been removed from the analysis. The rescaled cell perimeter represents the most probable cell perimeter for each condition, as determined from the normalized probability density function (PDF) of all measured cell perimeters. Distributions were normalized and plotted, and the cell perimeter corresponding to the peak of the PDF (maximum probability) was used for comparison across conditions. The number of cells per condition is summarized in Table EV1.

To build both Lewis' and Desh' law, data were separated with respect to their number of neighbors and binned in different classes before being averaged. To build Aboav-Weaire's law, the cells at the images' edge were removed, since their neighborhood is not defined. For that, the neighborhood of each cell is reconstructed to determine the average position of the collection of neighbors. If the distance between their average position and the central cells is less than 30 px, the central cell is assumed to be in the bulk of the image and not on the border of the image. To plot Aboav-Weaire's law, we computed $b(n)$ defined in (2).

$$b(n) = (n - 6)\mu_{m(n)} - \sigma_n^2 \qquad (2)$$

where $\mu_{m(n)}$ is the mean size of the neighborhood among the neighbors, and $\sigma_n^2$ is the central second moment of the distribution of neighbors, i.e., its variance. Such distribution is taken for a given mouse, not among different mice. Data from the different mice are average together with a ponderation proportional to the total number of cells making the statistics.

## Scanning electron microscopy (critical point drying)

After a week in culture on a polyacrylamide gel with the defined coating, iRPE were fixed in 3% glutaraldehyde (Agar scientific, Wetzlar, Germany) for 4 h at 4 °C followed by a washing in 0.1 M Soerensen's Phosphate Buffer (Merck, Darmstadt, Germany). The dehydrating step was performed in the ascending alcohol series with 10 min duration each. The samples were sequentially immersed in the following solutions: 30% ethanol, 50% ethanol, 70% ethanol, 90% ethanol, and three times in 100% ethanol.

Critical point drying was performed in liquid $CO_2$. Prior to imaging, cell monolayers were coated with 10 nm gold/palladium layer.

## Proliferation assay

Levels of proliferation in iRPE monolayers were assessed using a proliferation assay kit Click-iT EdU (Thermo Fisher, C10337). Cells were incubated with a 1:1 mixture of cell culture supernatant and 5-Ethinyl-2'-Desoxyuridin (EdU) (20 μM) containing iRPE cell culture medium for 24 h. Final EdU concentration in the mixture was 10 μM. Afterwards, cells were fixed as described for immuno-fluorescent labeling and processed according to the manufacturer's instructions. As EdU incorporates into newly synthesized DNA, proliferating cells were identified as having their nucleus positively stained with EdU-Alexa Fluor 488. The total area of the gel was imaged on Axio Observer 7 (Carl Zeiss) microscope as an overview using the tiles option. The ratio of proliferating cells was calculated in comparison with DAPI-positive nuclei and quantified using the particle analysis plugin of Fiji software (National Institutes of Health, USA).

## Photoreceptor outer segment internalization assay

Photoreceptor outer segments (POS) were isolated in batches from porcine eyes from a local slaughterhouse as previously described (Parinot et al, 2014). Briefly, the eyes were dissected using a red safelight lamp as the only illumination source, and the neural retina was collected. After retinal mechanical homogenization, POS were isolated via ultracentrifugation in a sucrose gradient. Thereafter, POS were labeled with 2.5 mg/ml FITC dye (Thermo Fisher, F1906), resuspended in DMEM (Thermo Fisher, 31331-028) containing 2.5% sucrose and stored in aliquots at −80 °C for future use. For internalization assay, POS aliquots were thawed in a water bath, washed with medium, centrifuged at $2300 \times g$ for 5 min and resuspended in iRPE cell medium to an amount of roughly 10 POS per iRPE cell in culture. Human MGF-E8 protein (R&D Systems, 2767-MF-050) and human protein-S (Coachrom, pp012A) were added at, respectively, 2.4 μg/ml and 2 μg/ml to aid the internalization. Next, iRPE monolayers were washed twice with warm media and incubated for 4 h at 37 °C with the prepared mixture at a density of approximately ten POS per cell. After incubation, the monolayers were gently washed four times with warm $PBS^{+/+}$ and fixed for fluorescence staining as described. Phagocytosis efficiency was calculated as projected area of internalized POS divided by the total projected POS area in the same field of view. Internalized POS were identified in orthogonal views from confocal optical sections (0.5 μm) by using F-actin staining to distinguish between apical-bound and internalized segments. Analyses were automatized with the particle analyses plugin of Fiji software (National Institutes of Health, USA) after applying a binary mask on the images.

## Traction force and monolayer stress microscopy

Traction force quantification and monolayer stress calculations were performed as previously described (Di Russo et al, 2021; Vishwakarma et al, 2018). Briefly, a week-old monolayer cultured on a PAA gel substrate containing fluorescent beads was imaged

using Axio Observer 7 (Carl Zeiss) microscope for the beads signal. Then, the monolayer was detached with 0.5% SDS solution in $PBS^{-/-}$ followed by washing steps and 20 min of gel incubation in cell media to prevent gel shrinkage due to the solvent change. The positions of the fluorescent beads in the hydrogel after the cell detachment (relaxed state of the gel) were compared with the tense gel before, and beads displacement vectors were calculated using the particle image velocimetry (PIV) plugin of Fiji (National Institutes of Health, USA). Traction forces were calculated from these vectors using the Fourier transform traction cytometry plugin (FTTC) of Fiji (National Institutes of Health, USA). Next, based on the traction force information, average normal stress vectors were calculated using a force balance algorithm in MATLAB (MathWorks) as formulated elsewhere (Di Russo et al, 2021; Vishwakarma et al, 2018). For each measured sample, the average cell height was calculated using a live actin staining (SirActin, Spirochrome) to correctly input the force balance algorithm.

## Stiffness measurements of iRPE monolayers

Measurements were performed using a Chiaro Nanoindenter System (Optics 11 Life) mounted on Zeiss Axio Observer 7 (Carl Zeiss) with an environmental chamber sustaining 37 °C. A probe of 10 μm radius and 0.025 N/m stiffness (Optics 11 Life) was used and the indentation was performed in Displacement control mode, indenting 10 μm into the cell monolayer. Approach curves were fitted via the DataViewer V2.4.0 software (Optics 11 Life) using a Hertzian contact model according to a set indentation depth from 0 to 3 μm for the apical stiffness. A Poisson's ratio of 0.5 was used and a maximum load of 30% was allowed for setting the contact point. Only results from well-fitted data ($R^2 > 0.95$) were considered for statistics.

## Drug experiments

For activation of the RhoA-dependent pathways, we incubated iRPE monolayers with Rho activator II (Cytoskeleton, CN03-A) and Rho kinase (ROCK) inhibitor Y-27632 (Sigma-Aldrich, Y0503) (Mao and Finnemann, 2021). Briefly, the 1-week-old iRPE monolayers on a hydrogel coated with high laminin density (20 μg/ml) were treated with Rho activator II (1 μg/ml) containing media for 3 h prior to POS phagocytosis assay or traction force measurements to promote the actomyosin contractility, or with an equal volume of water as a solvent control. To reduce the actomyosin contractility at low laminin surface density (2.5 μg/ml), ROCK inhibitor (25 μM) was added to the media 1 h prior to the POS phagocytosis assay or traction force measurements. An equal volume of DMSO was used as a solvent control. Further analysis was performed as described in the relevant chapters.

## Adhesion assay

For cell adhesion assay, a confluent monolayer of iRPE cells cultured in a 24-well plate was detached with 30 min incubation of 750 μl of 5 mM EDTA in $PBS^{-/-}$ without enzymatic treatment to prevent unwanted receptor cleavage. Cells were gently washed with an additional 750 μl of EDTA in $PBS^{-/-}$ and the whole solution was transferred into the 15 ml Falcon tube. After cell counting, the suspension was centrifuged at $300 \times g$ for 5 min. The pellet was resuspended in ice-cold media in half the volume required for the

experiment and transferred to low protein binding tubes. Ice-cold media should contain 20 μg/ml blocking antibody for blocking experiments. After 15 min of incubation on ice, the cell suspension was diluted (1:2) with the equivalent volume of warm media and immediately transferred onto the hydrogel surface. After 15 min at 37 °C and 5% $CO_2$, hydrogel surfaces were gently washed with warm $PBS^{+/+}$ to remove not attached cells, fixed with 2% PFA at room temperature for 8 min, and stained for F-Actin (Phalloidin-iFluor 488 Reagent, Abcam - ab176753) for analysis. The number of attached cells was quantified using the particle analysis plugin in Fiji software and normalized to the mean of control samples to convert as a percentage of adhesion. The following azide-free blocking antibodies were used: integrin β1 (clone P5D2) (Santa Cruz, sc-13590 L), integrin β4 (clone ASC-8) (Merk, MAB2059Z), integrin αvβ3 (clone LM609) (Merk, MAB1976Z), integrin α3 (clone P1B5) (Merk, MAB1952Z), integrin α5 (clone P1D6) (Merk, MAB1956Z), integrin α6 (clone GOH3) (Santa Cruz, sc-19622 L).

## Flow cytometry

One-week-old iRPE cells were carefully detached from the hydrogel substrate by incubation with 10 mM EDTA in $PBS^{-/-}$ at 37 °C and 5% $CO_2$ for 30 min, followed by Accutase solution (Sigma-Aldrich, A6964) for 15 min. During this period, gentle shaking was beneficial for detachment. Cells were gently washed from the surface with an additional 1 ml of EDTA in $PBS^{-/-}$, transferred into the 15 ml tube and centrifuged for 5 min at $300 \times g$. To minimize the impact of detachment on integrin surface localization, the procedure was performed rapidly on ice preventing integrin recycling. After centrifugation, cells were dispersed in ice-cold $PBS^{-/-}$ and divided into two separate tubes: one will contain the whole antibody panel, and another will be stained for β4 integrin alone. For the first tube, the viability staining with LIVE/DEAD™ Fixable Near IR(780) Viability Kit (Thermo Fisher Scientific, L34992) was performed during the 15-min incubation on ice protected from the light. After centrifugation at $300 \times g$ for 4 min, cells were fixed with 1 ml of 1% PFA for 5 min on ice and resuspended in 100 μl of $PBS^{-/-}$ containing 3% BSA and 5 mM EDTA (FACS buffer) for 6 min centrifugation at $300 \times g$. Fluorescently conjugated antibodies (see Reagents and Tools Table) were added according to the manufacturer's instructions and incubated with cells for 45 min on ice protected from the light. After washing, cells were resuspended in 400 μl of the FACS buffer and filtered using the Flowmi(R) Cell Strainer (MERK, BAH136800040) directly into the flow cytometry tube for the measurements. Stained cells were analyzed using a BD LSRFortessa flow cytometer equipped with 405, 488, 561, and 640 nm lasers. Analysis of the data was carried out using the FlowJo software.

## Statistics

Statistical analysis was performed using Prism (GraphPad). Data were tested for normality before proceeding to further statistical analyses. $p$-values were reported exactly unless they were below 0.0001, in which case they were reported as $p < 0.0001$. A $p$-value $\leq 0.05$ was considered statistically significant. Sample conditions and the specific test used for each data set are indicated in corresponding figure captions. Inclusion and exclusion criteria were pre-established prior to data collection. Samples (or animals) were excluded if they exhibited evident technical failure or did not meet pre-defined quality levels.

## Data availability

This study includes no data deposited in external repositories.

The source data of this paper are collected in the following database record: biostudies:S-SCDT-10_1038-S44319-025-00475-9.

## Peer review information

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

## Acknowledgements

We acknowledge the help of the Flow Cytometry Facility, a core facility of the Interdisciplinary Center for Clinical Research (IZKF) Aachen within the Faculty of Medicine at RWTH Aachen University. We appreciate the technical support of Adam Breitscheidel in the graphic design and Gloria's excellent help in imaging. A subset of the figures and text were included in the PhD thesis by Dr. Aleksandra Kozyrina (Kozyrina, 2024). This work in Di Russo's laboratory was supported by: A grant from the Interdisciplinary Centre for Clinical Research within the Faculty of Medicine at the RWTH Aachen University. Deutsche Forschungsgemeinschaft (DFG) grants (grants n° RU 2366/3-1 and GRK 2415/363055819). Add-on Fellowships of the Joachim Herz Foundation (ANK and TP).

## Author contributions

**Aleksandra N Kozyrina**: Data curation; Software; Formal analysis; Investigation; Visualization; Methodology; Writing—original draft; Writing—review and editing. **Teodora Piskova**: Data curation; Software; Formal analysis; Investigation; Visualization; Methodology; Writing—review and editing. **Francesca Semeraro**: Formal analysis; Investigation. **Iris C Doolaar**: Investigation; Methodology. **Taspia Prapty**: Software. **Tamás Haraszti**: Software; Formal analysis; Investigation; Methodology. **Maxime Hubert**: Data curation; Software; Formal analysis; Investigation; Methodology. **Reinhard Windoffer**: Data curation; Software; Formal analysis; Investigation; Methodology. **Rudolf E Leube**: Resources; Writing—review and editing. **Ana-Sunčana Smith**: Resources. **Jacopo Di Russo**: Conceptualization; Supervision; Funding acquisition; Visualization; Writing—original draft; Project administration; Writing—review and editing.

Source data underlying figure panels in this paper may have individual authorship assigned. Where available, figure panel/source data authorship is listed in the following database record: biostudies:S-SCDT-10_1038-S44319-025-00475-9.

## Funding

## Disclosure and competing interests statement

The authors declare no competing interests.

# Expanded View Figures

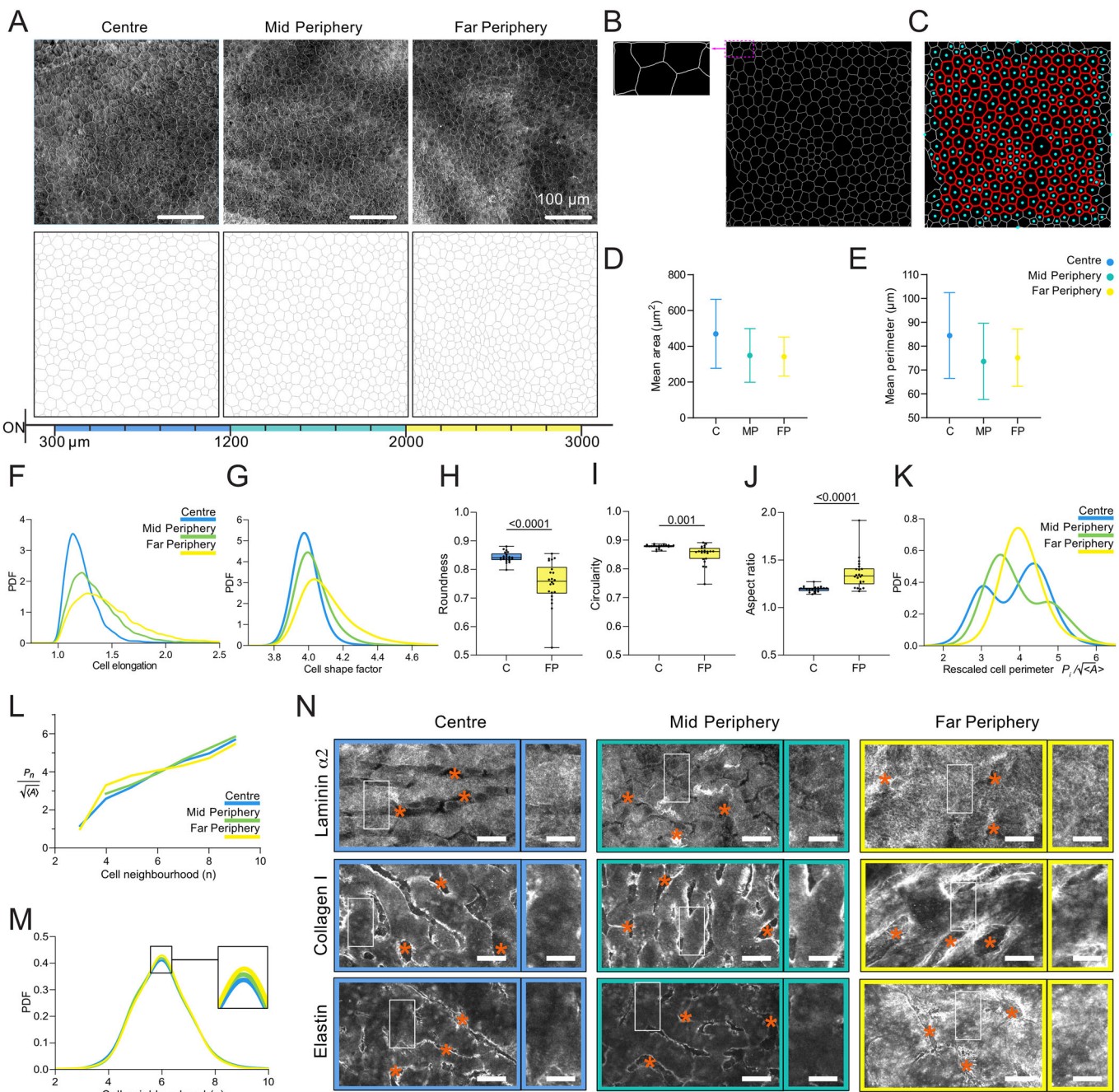

**Figure EV1. Murine RPE represents spatially different monolayer organization.**

(**A**) Representative confocal images of RPE stained for F-actin (top row) and post-processed binary images obtained by skeletonization used for the morphometric characterization upon segmentation (bottom row). Scale bar 100 µm. (**B**) Representative segmentation artefacts generated with Cellpose at the image edge. (**C**) The resulting cells (blue) and neighborhood (red) considered in the analyses. The average cellular area (**D**) and perimeter (**E**) shown with mean value and standard deviation for each region, respectively. Probability density function (PDF) of cell elongation (**F**) and cell shape factor (**G**) for different retinal regions. Nuclear geometric parameters such as roundness (**H**), circularity (**I**) and elliptical aspect ratio (**J**) show significant difference for cells from centre versus far periphery. The data was tested using Wilcoxon matched-pairs signed rank test; exact $p$-values are indicated on the figure for the respective comparisons. Box and whisker plots display the median (center line), 25th–75th percentiles (bounds of the box), and minimum to maximum values (whiskers). All individual data points are shown. (**K**) PDF of the rescaled cell perimeters defined as $P_i/\sqrt{<A>}$, $P_i$ being the individual cell perimeter. Topological characteristics of the RPE monolayer such as Desh's law (**L**) representing the relationship between cellular perimeter and shape, and PDF of cell neighborhood (**M**) varying for three different regions. More than 5500 cells (technical replicates) were analyzed from 3 mice (biological replicates). (**N**) Representative immunofluorescent images of the Bruch's membrane stained for laminin α2, collagen type I and elastin coupled with magnified view of quantified regions. Stars indicate preparation artefacts. Scale bar 20 µm and 10 µm for main images and magnifications, respectively.

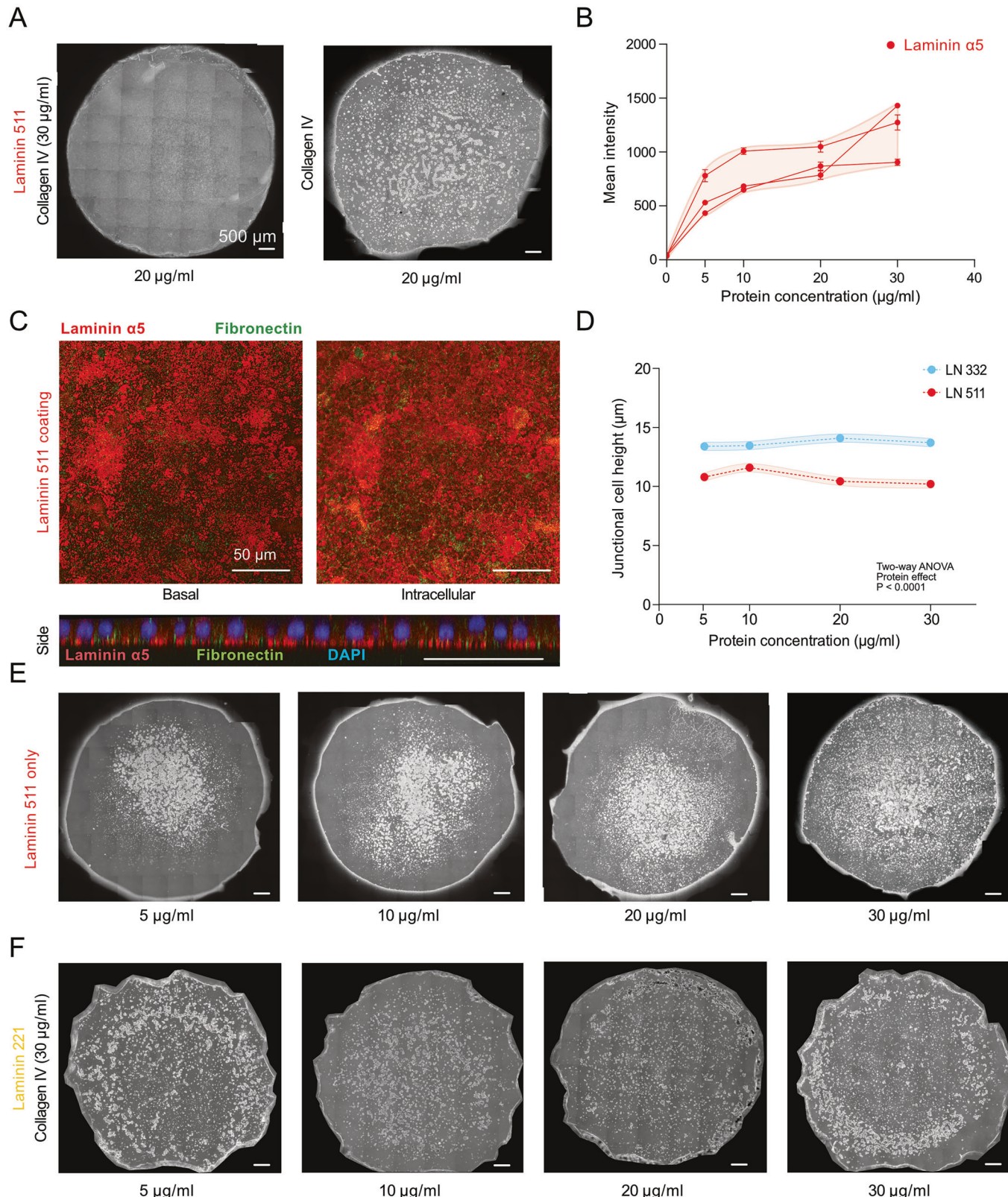

◀ **Figure EV2.  Protein composition of polyacrylamide gel surface coating is crucial for cell adhesion strength.**

(A) Representative image of complete iRPE monolayer obtained on 4 kPa polyacrylamide gel coated with laminin 511 (20 μg/ml) combined with collagen type IV (30 μg/ml) in contrast with the fragmented monolayer on pure collagen type IV (20 μg/ml). Scale bar 500 μm. (B) Line graph of the mean fluorescent intensity of the hydrogel surface fixed and stained with laminin α5 antibody right after chemical crosslinking with different concentrations of laminin 511 combined with collagen type IV (30 μg/ml). This relation indicates correlation between laminin concentration and surface density. Each line represents individual dilution series. Images were taken from 10 different areas of the gel (technical replicates) from 3 independent experiments (biological replicates). Data are shown as mean within the gel ± SEM. (C) Immunofluorescent images of the laminin 511 coated gel surface stained for ECM proteins show the absence of fibronectin protein deposition after one week in cellular culture. Top row represents the signal from laminin α5 (red) and fibronectin (green) at the basal surface and within the monolayer. Fibronectin signal appears to be intracellular as shown in the side view of the monolayer. (D) Average cell height measured at the adherence junction levels of one-week-old iRPE monolayer. Measurements are the mean values from three fields of view (technical replicates) from at least three independent experiments (biological replicates) ± SEM. Statistic was performed using a mixed-effects model (REML) with the Geisser-Greenhouse correction, Tukey's multiple comparisons test. (E, F) Representative overview images of colonies iRPE cells seeded on gels coated with different concentrations of laminin 511 and laminin 221 combined with collagen type IV (30 μg/ml), indicating the insufficient cellular adhesion and monolayer formation in contrast to standard experimental conditions. Scale bar 500 μm.

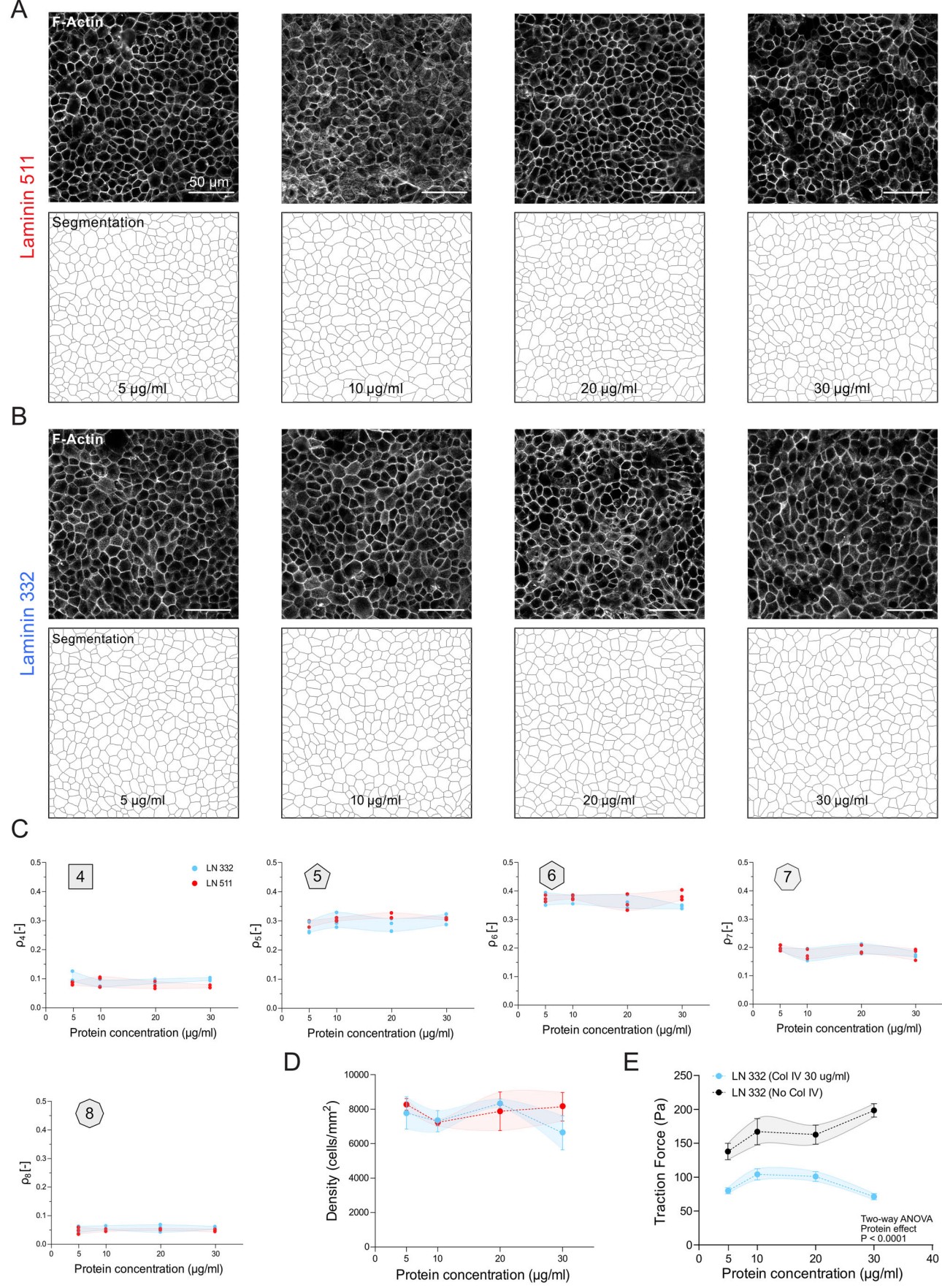

◀ **Figure EV3. Structure and topological stability of iRPE in vitro.**

Representative confocal images of one-week cultured iRPE monolayers on 4 kPa PAA gels coated with different concentrations of laminin 511 (A) or laminin 332 (B) both in presence of collagen type IV (30 µg/ml). The top row represents the actin cytoskeleton network in the stained samples and the bottom row shows segmented images used for morphometric analysis. (C) Probabilities of the cellular neighborhood ($\rho_n$) from 4 ($\rho_4$) "squares" to 8 ($\rho_8$) "octagon" indicating the stable neighborhood environment regardless of the protein surface concentration. Data are from 3 independent experiments (biological replicates) with 3 independent fields of view in each experiment (technical replicates). The total number of analyzed single cells is shown in Table EV1. (D) Quantification of cellular density across different experimental conditions shown as average between 3 biological replicates ± SEM. (E) The presence of collagen type IV in the laminin 332 containing coating solution decreases the traction forces exerted by cells. This difference becomes less significant at lower laminin concentration. Data are shown as mean of at least 4 biological replicates each with at least 6 technical replicates ± SEM. The statistical significance was assessed using mixed-effects model (REML) with the Geisser-Greenhouse correction and Tukey's multiple comparisons test.

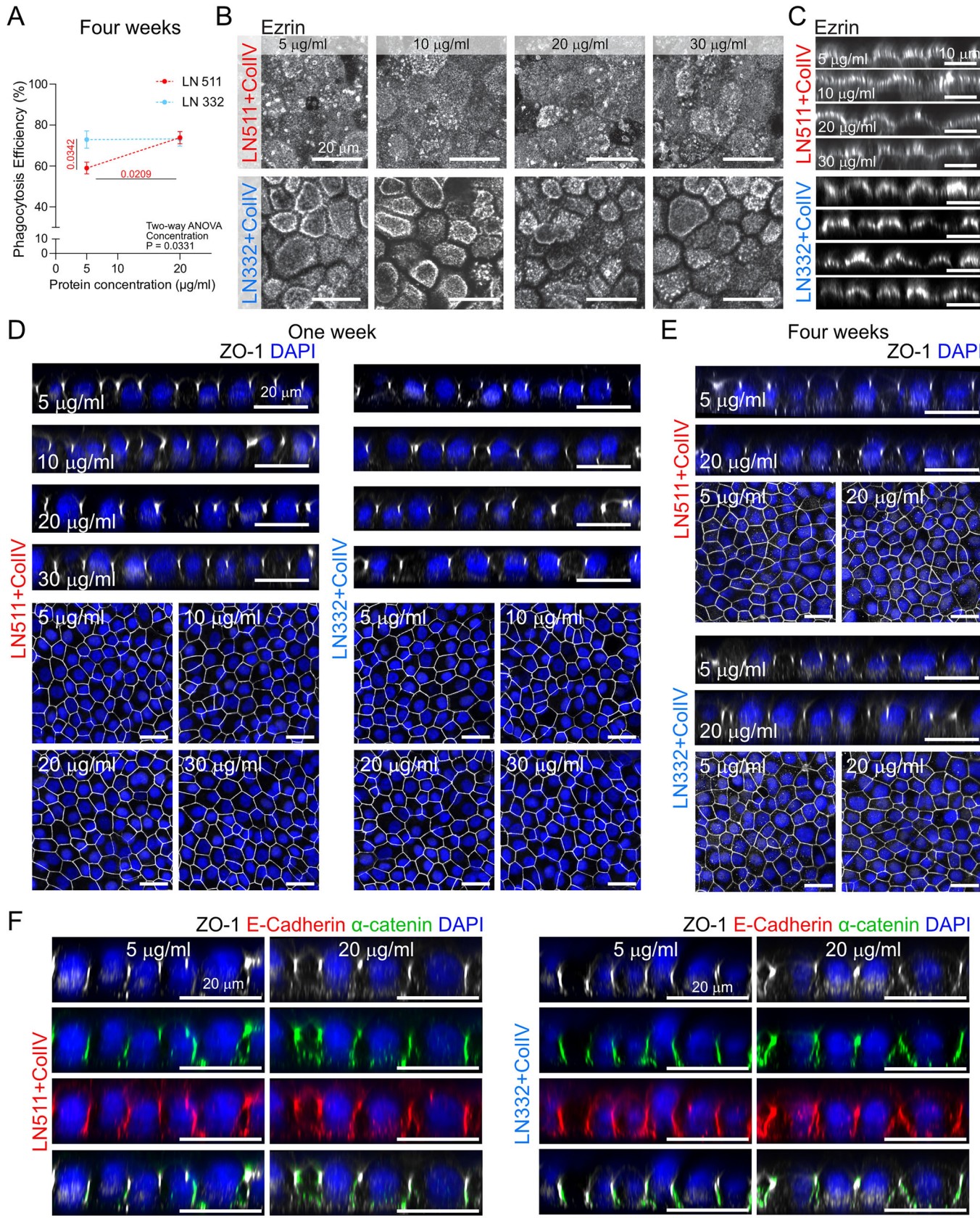

**Figure EV4.   Effect of laminin coating on iRPE cellular maturity, polarization, and functionality.**

(A) Quantification of phagocytosis efficiency shown as an average between at least 3 technical replicates± SEM in 4-week-old iRPE cultures from 4 independent experiments (biological replicates), comparing different laminin isoforms and coating concentrations. The data were statistically tested using two-way ANOVA with Tukey's multiple comparison test. Exact *p* values are illustrated on the respective comparisons, with red values referring to multiple comparison test results. Representative Ezrin immunostaining overview (B) and orthogonal projections (C), representing microvilli organization across all laminin isoforms and concentrations. Orthogonal projections of tight junctions (ZO-1) and DAPI-stained nuclei, with XY plane images, for iRPE cultures cultured on different laminin isoforms and densities for one week (D) and four weeks (E). (F) Representative orthogonal projections showing localization of ZO-1, E-cadherin, alpha-catenin, and DAPI across both laminin isoforms at low and high coating densities.

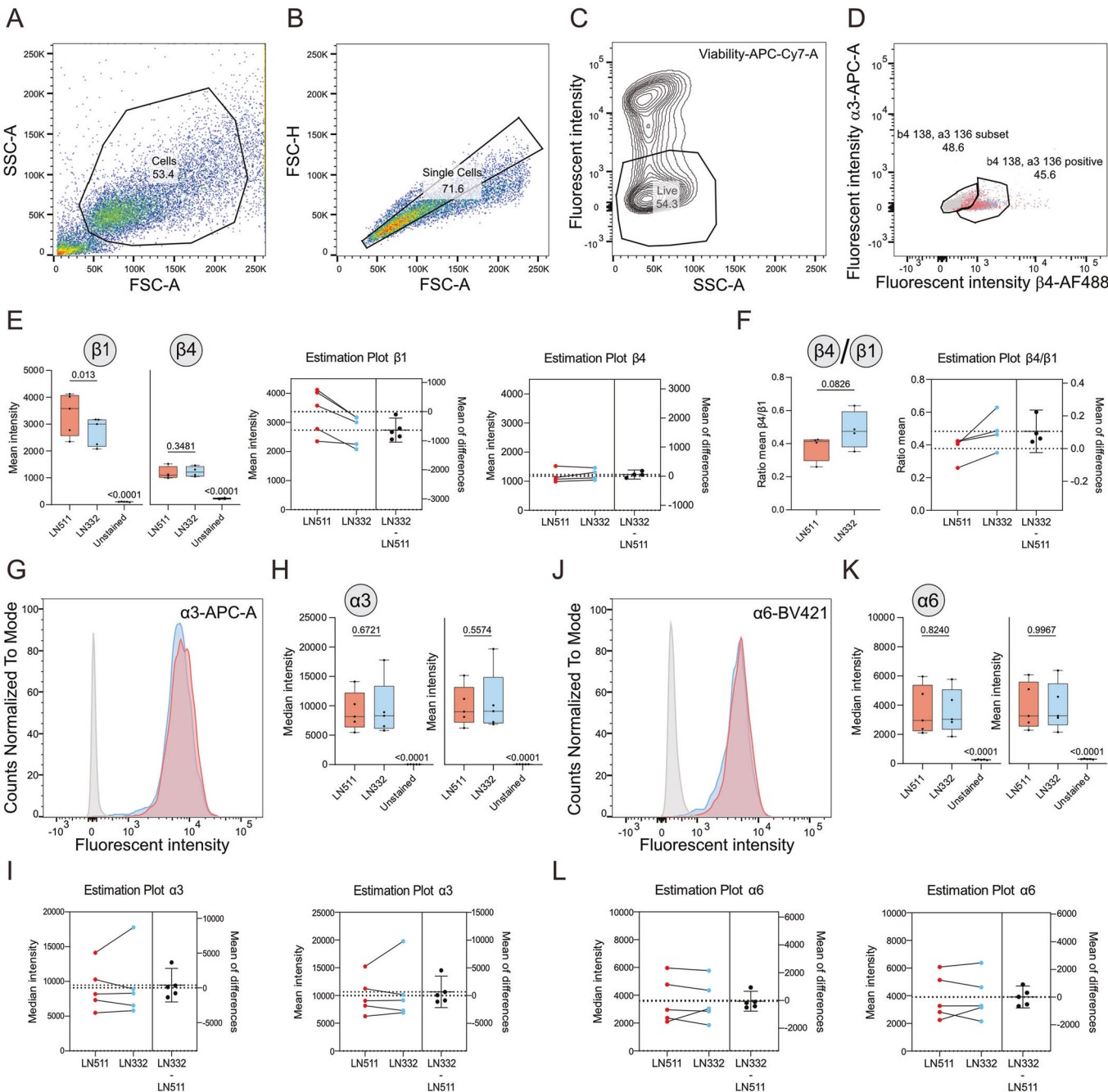

**Figure EV5. Flow cytometry analysis of iRPE cultured on polyacrylamide gels coated with 5 µg/ml of laminin 511 or laminin 332.**

Representative dot plots showing the gating strategy to identify cells (**A**), single cells (**B**), contour plot to extract a population of single living cells (**C**) and gating strategy for β4-positive cells (**D**). (**E**) Box and whisker plots of the mean intensity of integrin β1 and integrin β4 staining (left) and corresponding estimation plots (right) acquired from flow cytometry data. (**F**) Box and whisker (left) and estimation (right) plots showing the ratio between mean intensity of integrin β4 and integrin β1 staining. Representative histograms of flow cytometry analysis, box and whisker plots of median and mean fluorescent intensity and corresponding estimation plots of staining for APC-A-conjugated integrin α3 (**G–I**) and BV421-conjugated integrin α6 (**J–L**). The data was obtained from 5 independent experiments (biological replicates). Box and whisker plots display the median (center line), 25th–75th percentiles (bounds of the box), and minimum to maximum values (whiskers). All individual data points are shown. The gray-colored histogram represents the signal from unstained cells. Statistical analysis was done using a paired t-test to compare results between laminin 511 and laminin 332 coatings and one-way ANOVA with Dunnett's multiple comparison test for comparison with unstained sample. Exact *p* values are illustrated on the respective comparisons.

