## [Peer Review File · EMBO Reports]

Laminin-defined Mechanical Status Modulates Retinal Pigment Epithelium Phagocytosis

Aleksandra Kozyrina, Teodora Piskova, Francesca Semeraro, Iris Doolaar, Taspia Prapty, Tamas Haraszti, Maxime Hubert, Reinhard Windoffer, Rudolf Leube, Ana-Suncana Smith, and Jacopo Di Russo

Corresponding author(s): Jacopo Di Russo (jdirusso@ukaachen.de)

Review Timeline:

Transfer Date:	13th Mar 25
Editorial Decision:	2nd Apr 25
Revision Received:	7th Apr 25
Accepted:	29th Apr 25

Editor: Deniz Senyilmaz Tiebe

Transaction Report: This manuscript was transferred to EMBO reports following peer review at The EMBO Journal.

Referee #1:

Report:

This is a revised manuscript by Kozyrina. et al. demonstrating that different isoforms and density of laminin generate specific biochemical signals that affect RPE contractility and phagocytic capacity. This revised ms offers greater clarity in experimental design choices and strengthens the rigor of its conclusions by incorporating additional experiments. However, one part of the main conclusion remains to be clarified.

Since the authors now refined their focus of laminin's broad effect on RPE function to its specific impact on phagocytosis, I believe a more nuanced discussion on this effect is warranted. In the MS, the authors evaluated RPE's phagocytic capacity using a % Phagocytic Efficiency, as calculated by internalized POS-FITC (below actin belt) divided by total POS-FITC after a 4-hour continuous incubation of POS-FITC. In most RPE models, this timeline is long enough that both POS internalization and degradation have occurred. Therefore, the calculated phagocytic efficiency is actually affected by both the rate of internalization, and the rate of degradation. Since the author called it the "POS internalization assay", I assume here that when describing the results, the authors were mainly concerned about the rate of internalization, assuming the rate of degradation was not affected. However, without adopting a more direct assessment as described in Hazim et al 2017 (DOI: 10.1186/s13287-017-0652-9), the authors cannot really conclude an increased or decreased RPE phagocytic capacity under conditions investigated here.

Examples:

1. Abstract: ...which subsequently diminishes phagocytic capacity.
2. Line 246-248
3. Line 273-274

Can the authors either include additional experiments to delineate the effect of ingestion vs. degradation rate in their system or discuss the caveat of using "Phagocytic Efficiency" to assess phagocytic capacity?

Referee #3:

The reviewer read the response to the three reviewer's concerns.

Although the authors considered all of them, their response is differently acceptable.

Authors provide some of the requested experiments as in vivo stainings, however these turned to not give trends going in the direction of the in vivo experiments as no major differences were detected in function of the regions analysed. Basically, the authors say that there is not enough resolution. However, this is just qualitative and they did not apply any real quantitative analysis, so we do not know for example in laminin and integrins are differentially expressed.

Some others answers like for the choice of PAA stiffnesses failed to explain why stiffnesses similar to in vivo were not used.

On the other side the authors did not mitigate their conclusions, which would be in line with the concerns raised by the reviewers and that have not been fully erased in the revision process. This work remains an important experimental tour de force that however remains a puzzle with too many holes. It could be published if authors clearly recognize this fact and tone down their conclusions.

Dear Jacopo,

Thank you for transferring your manuscript, which was previously revised and re-reviewed at The EMBO Journal. As previously communicated in my colleague leva's previous decision letter, we would like to offer publication pending a satisfactory minor (textual) revision, where the emphasis on phagocytosis is toned down and the caveats with the experimental design indicated by referee #1 is carefully discussed. Moreover, the conclusions drawn from in vivo assays should also be adjusted as pointed out by referee #3 and the shortcomings mentioned should be acknowledged in the manuscript text.

Moreover, I need you to address the points below before I can accept the manuscript.

- Please provide source data as requested by our Source Data Coordinator Dr. Hannah Sonntag on 21.03.2025.
- During our routine analyses, we noted that there are textual overlaps with a document, which we realize is the PhD dissertation of the first author Dr. Aleksandra Kozyrina (<https://d-nb.info/1352014297/34>). Given that the dissertation is formally published, we would like to ask you to include formal citation in the Acknowledgements section with a sentence along these lines: "A subset of the figures and text were included in the PhD thesis by Dr. Aleksandra Kozyrina (citation) and also to include the citation in the Reference list.
- Please provide the manuscript in word format without the figures (but with the legends).
- Please rename the "Data and materials availability" section as "Data Availability". As per our format requirements, please replace the statement with the following: "This study includes no data deposited in external repositories."
- Please rename the "Competing interests" section as "Disclosure And Competing Interests Statement".
- Please remove the "Author Contributions" section from the manuscript text.
- Please fill out and include an author checklist as listed in our online guidelines (<https://www.embopress.org/page/journal/14693178/authorguide>).
- Please make sure that the funding information is also complete in the manuscript tracking system, which is currently missing. Moreover, the funding information provided in the manuscript needs to be part of the Acknowledgments section, so the separate section with the heading Funding needs to be removed.
- All research articles submitted as revised versions must include a structured methods section that includes a Reagents and Tools Table followed by a Methods and Protocols section. Please see <https://www.embopress.org/page/journal/14693178/authorguide#structuredmethods> for further information.
- Along similar lines, we note that Table 1 needs to be a part of the Reagents and Tools Table.
- We note that Table 2 is size-wise suitable to be included into the manuscript text. If you would like to keep it as a separate file, the nomenclature should be Table EV1; in which case, the manuscript callouts should be updated accordingly.
- The nomenclature of Figures S1-S5 should be corrected as Figure EV1, Figure EV2, etc. Their source file name, title, legends and manuscript callouts need to be updated accordingly.
- The manuscript sections should be in the following order: Title page - Abstract & Keywords - Introduction - Results - Discussion - Methods - Data Availability - Acknowledgments - Disclosure Statement & Competing Interests - References - Figure Legends - (Main Tables with legends if applicable) - Expanded View Figure Legends.
- "Materials and Methods" section needs to be renamed as "Methods".
- During our routine figure checks, we noted a potential image re-use between Figure 3C and 3J. Please clarify.
- Our production/data editors have asked you to clarify several points in the figure legends - Figure Legends (main + EV):
 - o Please note that the exact p values are not provided in the legends of figures 1C, G, H, J; 2I, L; 3E, F, H, I; 4A-E; 5B, D, G, H, I; S3 E.
 - o Please indicate what */ **/ ***/ **** represents; if this represents p value(s), please indicate the statistical test used and where appropriate, specify the exact p value in the legend(s) of figure(s) S1 H-J
 - o Please note that the box plots need to be defined in terms of minima, maxima, centre, bounds of box and whiskers, and percentile in the legends of figures 2D, 4A-E; S1H-J
 - o Please note that information related to n is missing in the legends of figures 3E, F, H, I; 4A-E; 5B, D; S1 D, E, H-J; S2 B, S4 A, S5 D, E, F, J.
 - o Please note that the error bars are not defined in the legends of figures 5B, D; S4 A
- Papers published in EMBO Reports include a 'synopsis' and 'bullet points' to further enhance discoverability. Both are displayed on the html version of the paper and are freely accessible to all readers. The synopsis includes a short standfirst summarizing the study in 1 or 2 sentences (max 35 words) that summarize the paper and are provided by the authors and streamlined by the handling editor. I would therefore ask you to include your synopsis blurb and 3-5 bullet points listing the key experimental findings.
- In addition, please provide an image for the synopsis. This image should provide a rapid overview of the question addressed in the study but still needs to be kept fairly modest since the image size cannot exceed 550 (width) x 300-600 (height) pixels.

Thank you again for giving us to consider your manuscript for EMBO Reports, I look forward to your minor revision.

Kind regards,

Deniz

--

Deniz Senyilmaz Tiebe, PhD
Senior Scientific Editor
EMBO Reports

All editorial and formatting issues were resolved by the authors.

Dr. Jacopo Di Russo
RWTH Aachen University
Wendingweg 2
Aachen, NRW 52074
Germany

Dear Jacopo,

Thank you for submitting your revised manuscript. I have now looked at everything and all is fine. Therefore, I am very pleased to accept your manuscript for publication in EMBO Reports.

Congratulations on a nice work!

Kind regards,

Deniz

--

Deniz Senyilmaz Tiebe, PhD
Senior Scientific Editor
EMBO Reports

--
